# Phylogenetic analysis of 17271 Indian SARS-CoV-2 genomes to identify temporal and spatial hotspot mutations

**Nimisha Ghosh**[1,2], **Suman Nandi**[3], **Indrajit Saha**[3] *

1 Faculty of Mathematics, Informatics and Mechanics, University of Warsaw, Warsaw, Poland, 2 Department of Computer Science and Information Technology, Institute of Technical Education and Research, Siksha 'O' Anusandhan (Deemed to be University), Bhubaneswar, Odisha, India, 3 Department of Computer Science and Engineering, National Institute of Technical Teachers' Training and Research, Kolkata, West Bengal, India

☯ These authors contributed equally to this work.
* indrajit@nitttrkol.ac.in

**Data Availability Statement:** All relevant data are available on Figshare: https://doi.org/10.6084/m9.figshare.19328735.v1.

**Funding:** This work was carried out during the tenure of an ERCIM 'Alain Bensoussan' Fellowship

## Abstract

The second wave of SARS-CoV-2 has hit India hard and though the vaccination drive has started, moderate number of COVID affected patients is still present in the country, thereby leading to the analysis of the evolving virus strains. In this regard, multiple sequence alignment of 17271 Indian SARS-CoV-2 sequences is performed using MAFFT followed by their phylogenetic analysis using Nextstrain. Subsequently, mutation points as SNPs are identified by Nextstrain. Thereafter, from the aligned sequences temporal and spatial analysis are carried out to identify top 10 hotspot mutations in the coding regions based on entropy. Finally, to judge the functional characteristics of all the non-synonymous hotspot mutations, their changes in proteins are evaluated as biological functions considering the sequences by using PolyPhen-2 while I-Mutant 2.0 evaluates their structural stability. For both temporal and spatial analysis, there are 21 non-synonymous hotspot mutations which are unstable and damaging.

## Introduction

It is now close to two years since the emergence of SARS-CoV-2, the virus behind the deadly COVID-19 disease and the scientific community is still struggling to put an end to this pandemic. Though India was able to contain the spread in the first wave, the second wave put the entire system in turmoil. In September 2021, around 30,000 https://www.covid19india.org/ cases were being registered on a daily basis while in the month of May, this figure surpassed 300,000. Scientists and researchers had attributed this surge due to the evolution of this contagious virus which has resulted in Delta (B.1.617.2) variant. Though the vaccination drive in India is in full swing, doubts regarding the efficacy of the vaccine against such mutations cannot be undermined. Apart from Delta, other variants of concern as declared by W.H.O making their rounds are Alpha (B.1.1.7) [1], Beta (B.1.351) [2] and Gamma (P.1) [3] variants. All these

Program awarded to Dr. Nimisha Ghosh. This work has also been partially supported by CRG short term research grant on COVID-19 (CVD/2020/000991) from Science and Engineering Research Board (SERB), Department of Science and Technology, Govt. of India. However, it does not provide any publication fees.

**Competing interests:** The authors have declared that no competing interests exist.

variants, especially Delta resulted in new spurts of lockdown in the country. Thus, to understand its frequent mutations, a study pertaining to the evolution of SARS-CoV-2 virus is inevitable [4, 5].

To understand these evolutionary mutations, 103 SARS-CoV-2 sequences have been analysed by Tang et al. [6] which revealed two major lineages, L and S. These lineages are defined by two tightly linked SNPs at positions at 28144 (ORF8: C251T, S84L) and 8782 (orf1ab: T8517C, synonymous) and might influence virus pathogenesis. Raghav et al. [7] have used RTIC primers–based amplicon sequencing to profile 225 Indian SARS-CoV-2 sequences. Their analysis showed that apart from local transmission, Europe and Southeast Asia are the two major routes for introduction of the disease in India. Their study also revealed that D614G in the Spike protein as a very common mutation that increases virus shedding and infectivity. In [8], Wang et al. have proposed a h-index mutation ratio criteria to evaluate the non-conserved and conserved proteins with the help of over 15K sequences. As a result, Nucleocapsid, Spike and Papain-like protease are found to be highly non-conserved while Envelope, main protease, and Endoribonuclease protein are considered to be conservative. They have further identified mutations on 40% of nucleotides in Nucleocapsid gene, thereby reducing the efforts on the ongoing development of various COVID-19 diagnosis and cure which targets Nucleocapsid gene. Similar analysis conducted by Yuan et al. [9] with 11183 sequences revealed 119 high frequency substitutions as SNPs around the globe. Among the nucleotide changes in SNPs, C to T is the major one which indicates adaptation and evolution of the virus in the human host which can pose new challenges. Also, they have found Nucleocapsid to have the highest mutational changes in frequency. Thus both the works by Wang et al. [8] and Yuan et al. [9] refute the claim by Ascoli [10] that Nucleocapsid can be a possible diagnostic target. Thus, it is important to understand the evolution of SARS-CoV-2 over time. Cheng et al. [11] have identified five major mutation points such as C28144T, C14408T, A23403G, T8782C and C3037T in almost all strains for the month of April 2020. Their functional analysis show that these mutations lead to a decrease in protein stability and eventually a reduction in the virulence of SARS-CoV-2 while A23403G mutation increases the Spike-ACE2 interaction leading to an increase in its infectivity. Phylogenetic analysis done by Maitra et al. [12] shows that mutations such as C14408T in RdRp and A23403G in Spike majorly encompass A2a clade in 9 Indian sequences. Moreover, a triplet based mutation such as 2881–3 GGG/AAC in Nucleocapsid gene which might be responsible for affecting miRNAs bindings to original sequences has also been reported in their work. Guruprasad et al. [13] has analysed 10333 spike protein sequences out of which 8155 proteins comprised of one or more mutations, leading to a total of 9654 mutations that correspond to 400 distinct mutation sites. According to this analysis the top 10 mutations according to the total number of occurrences are D614 (7859), L5 (109), L54 (105), P1263 (61), P681 (51), S477 (47), T859 (30), S221 (28), V483 (28) and A845 (24). Other important works like [14–17] have also revealed different mutations after analysis of several SARS-CoV-2 sequences. Looking at these varied mutations as reported by all the aforementioned works, it can be easily concluded that the evolutionary study of SARS-CoV-2 genomes is very relevant in the current pandemic scenario of the ongoing waves in India.

Motivated by the aforementioned studies, in this work we have performed multiple sequence alignment (MSA) of 17271 Indian SARS-CoV-2 genomes using multiple alignment using fast fourier transform (MAFFT) [18] followed by their phylogenetic analysis using Nextstrain [19] to eventually identify hotspot mutations both month-wise (temporal) and state-wise (spatial). Thereafter, from the aligned sequences, temporal and spatial analysis are carried out to identify top 10 hotspot mutations in the coding regions based on entropy, thereby resulting in 130 and 250 hotspot mutations respectively. Finally, to judge the functional characteristics of all the non-synonymous hotspot mutations, their changes in proteins are

evaluated as biological functions considering the sequences by using PolyPhen-2 while I-Mutant 2.0 evaluates their structural stability. The hotspot mutations which are unstable and damaging and common in both the categories are T77A and V149A in NSP6, T95I and E484Q in Spike, Q57H and T223I in ORF3a, I82S and I82T in Membrane, D119V and F120L in ORF8, R203K, R203M and G215C in Nucleocapsid. Furthermore, as recognised by virologists, E484K in Spike which is identified in temporal analysis is yet another major mutation which is responsible for improving the ability of the virus to escape the host's immune system [20].

## Material and methods

In this section, the dataset collection for the 17271 Indian SARS-CoV-2 genomes are discussed along with the proposed pipeline.

### Data acquisition

To perform the multiple sequence alignment and phylogenetic analysis, 17271 Indian SARS-CoV-2 genomes are collected from Global Initiative on Sharing All Influenza Data (GISAID) https://www.gisaid.org/ and the Reference Genome (NC 045512.2) https://www.ncbi.nlm.nih.gov/nuccore/1798174254 is collected from National Center for Biotechnology Information (NCBI). The SARS-CoV-2 sequences are mostly distributed from January 2020 to September 2021 across the states of India. Moreover, for mapping the protein sequences and the subsequent changes in the amino acid, protein PDB are collected from Zhang Lab https://zhanglab.ccmb.med.umich.edu/COVID-19/. These PDBs are then used to model and identify the structural changes in the protein. All these analyses are performed on High Performance Computing facility of NITTTR, Kolkata while MATLAB R2019b is used for checking the amino acid changes.

### Pipeline of the work

The pipeline of the work is provided in Fig 1. Initially, multiple sequence alignment (MSA) of 17271 Indian SARS-CoV-2 genomes is performed using MAFFT which is followed by their phylogenetic analysis using Nextstrain, thereby leading to the identification of mutation points as SNPs. In this work, MAFFT is used as the MSA tool. As MAFFT uses fast fourier transform thus, it scores over other alignment techniques. So, MAFFT is used in this work for MSA. On the other hand, by taking the advantage of Nextstrain, in this work the evolution and geographic distribution of SARS-CoV-2 genomes are visualised by creating the metadata in our High Performance Computing environment.

Once the alignment and the phylogenetic analyses are completed and the mutation points as SNPs are identified, temporal (month-wise) and spatial (state-wise) analysis are performed for the aligned sequences to identify top 10 hotspot mutations both month-wise and state-wise. Furthermore, amino acid changes in the SARS-CoV-2 proteins are also identified considering the codon table. The top 10 hotspot mutations are identified for each month and each state based on their entropy values for the coding regions and are computed as follows:

$$\mathcal{E} = ln\ 5 + \sum \theta_\alpha^\beta\ \left[\ ln\ (\theta_\alpha^\beta)\ \right] \tag{1}$$

where $\theta_\alpha^\beta$ represents the frequency of each residue $\alpha$ occurring at position $\beta$ and 5 represents the four possible residues as nucleotides plus gap. Subsequently, the amino acid changes for the temporal and spatial non-synonymous hotspot mutations are visualised graphically. Finally, the amino acid changes of the non-synonymous hotspot mutations are considered to evaluate their functional characteristics and they are visualised in the respective protein structure as well.

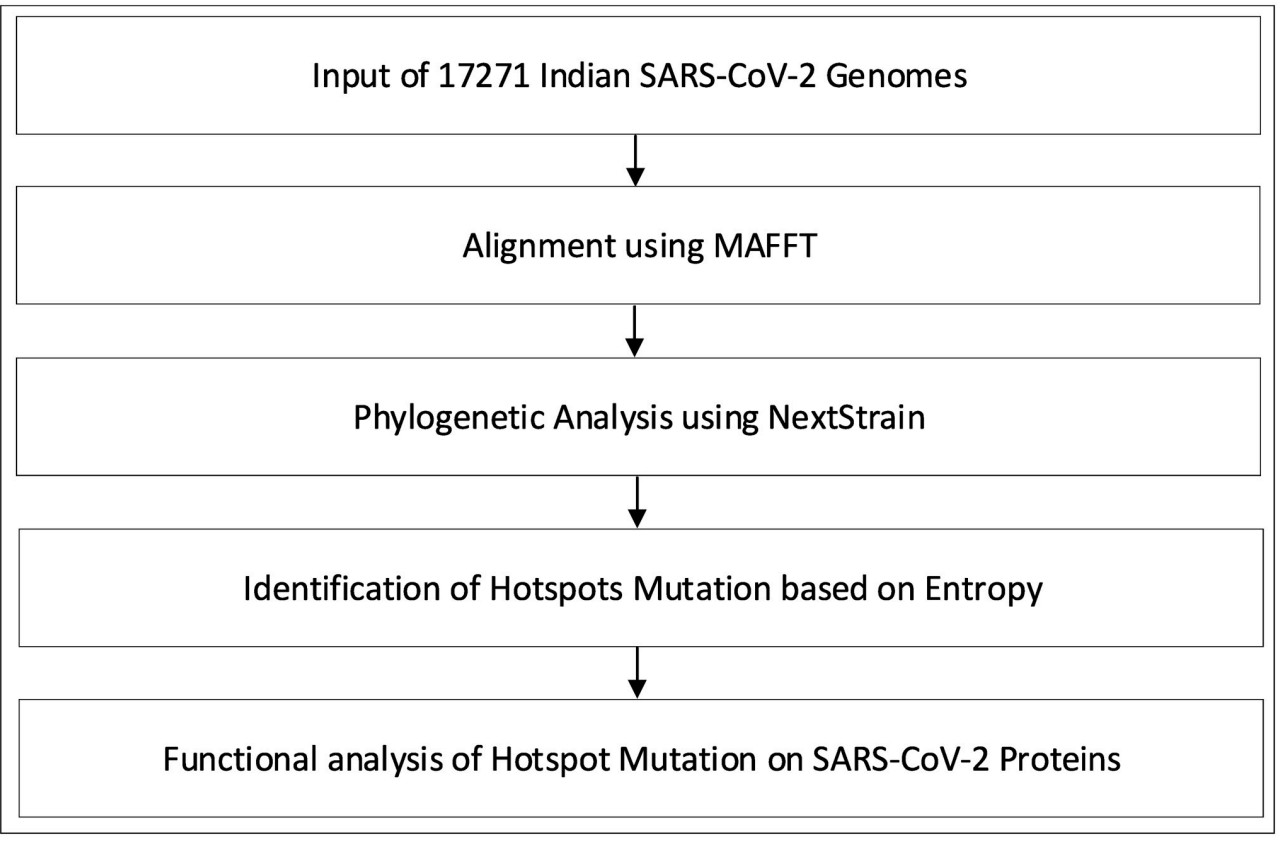

**Fig 1. Pipeline of the work.**

## Results

All the experiments in this work are carried out according to Fig 1. In this regard, MSA of 17271 Indian SARS-CoV-2 genomes is initially carried out using MAFFT. Thereafter, their phylogenetic analysis using Nextstrain reveals 5 virus clades viz. 19A, 19B, 20A, 20B and 20C and also the corresponding mutation points as SNPs. Subsequently, temporal (month-wise) and spatial (state-wise) analysis are performed for the aligned sequences to identify the top 10 hotspot mutations in each category, resulting in 190 and 250 mutation points respectively. The phylogenetic trees in radial and rectangular views considering temporal analysis are shown in Fig 2(a) and 2(b) while Fig 2(c) and 2(d) show the views considering spatial analysis. The normal and zoomed views of the geographical distribution of the sequences clade-wise are shown in Fig 2(e) and 2(f) respectively. In unsupervised learning feature selection is a non-trivial task; entropy of the aligned sequences is considered to be the selected feature in this work. For example, temporal analysis of January-March-2020 with 191 sequences shows that G11083T in NSP6 has the highest entropy value of 0.82391 while for spatial analysis of Maharastra with 3674 sequences, the highest entropy value of 1.02173 is borne by G28881A and G28881T in Nucleocapsid. Such results are reported in Tables 1 and 2 for the top 10 hotspot mutations for temporal and spatial analysis along with the associated details while S1 and S2 Tables in S1 File report the list of all temporal and spatial hotspot mutations. Table 2 reports the spatial analysis for the states of India. The entropy values corresponding to the nucleotide changes are shown in Fig 2(g) while the temporal and spatial changes in entropy are reported in S3 and S4 Tables

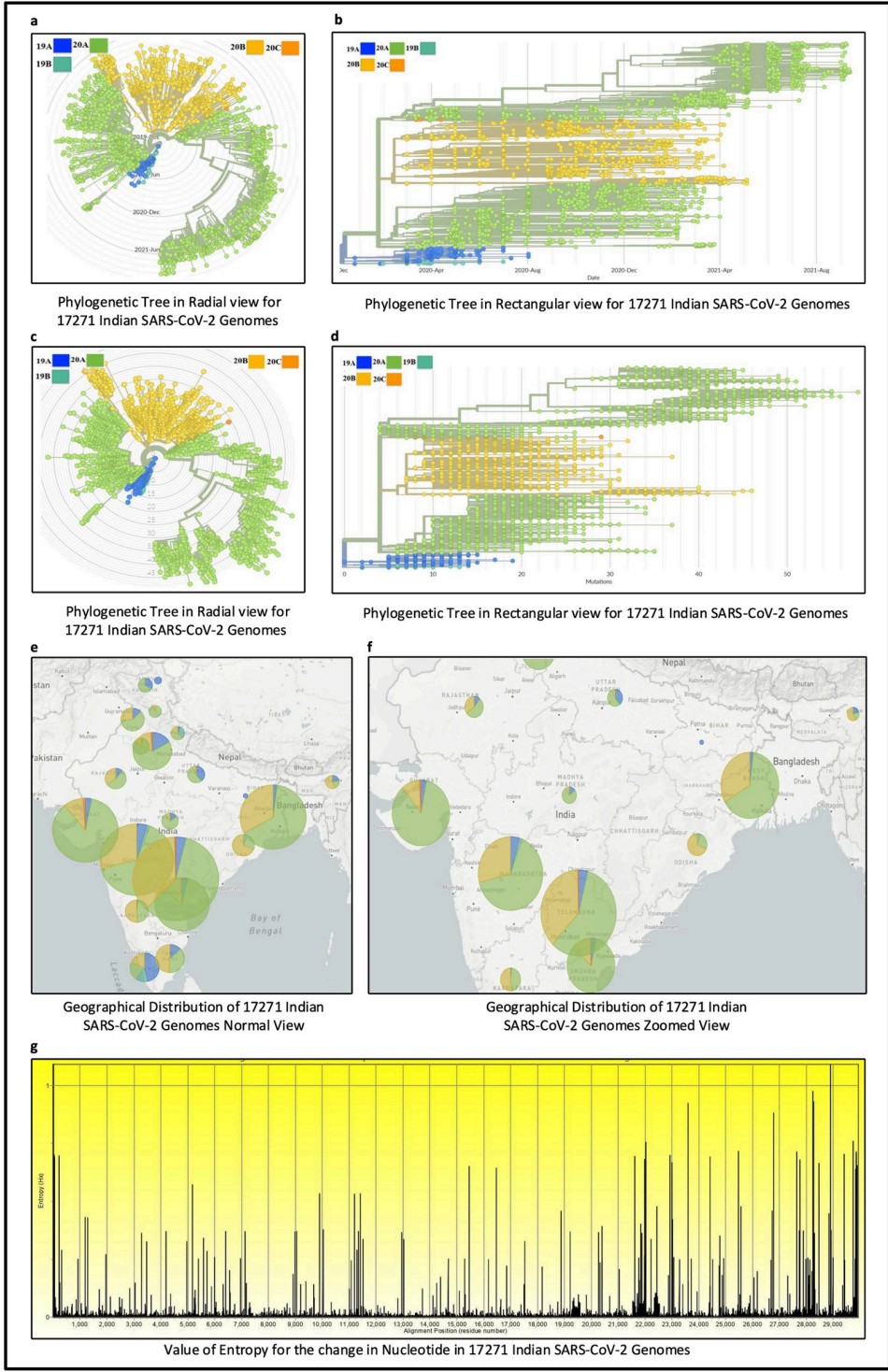

**Fig 2.** Phylogenetic analysis of 17271 Indian SARS-CoV-2 Genomes where (a) and (b) show the phylogenetic tree in radial and rectangular views for 17271 Indian SARS-CoV-2 genomes for temporal analysis, (c) and (d) show the phylogenetic tree in radial and rectangular views for 17271 Indian SARS-CoV-2 genomes for spatial analysis, (e) and (f) are the geographical distribution in normal and zoomed views and (g) shows the value of entropy for the change in nucleotide.

**Table 1. List of top 10 hotspot mutations based on temporal analysis.**

| Month | Number of Sequences | Genomic Coordinate | Entropy | Nucleotide Change | Amino Acid Change | Protein Coordinate | Coding Region |
|---|---|---|---|---|---|---|---|
| January-March-2020 | 191 | 11083 | 0.82391 | G>T | L>F | 37 | NSP6 |
| | | 28311 | 0.64212 | C>T | P>L | 13 | Nucleocapsid |
| | | 3037 | 0.63531 | C>T | F>F | 106 | NSP3 |
| | | 14408 | 0.63531 | C>T | P>L | 323 | RdRp |
| | | 23403 | 0.63531 | A>G | D>G | 614 | Spike |
| | | 23929 | 0.63088 | C>T | Y>Y | 789 | Spike |
| | | 6312 | 0.59276 | C>A | T>K | 1198 | NSP3 |
| | | 13730 | 0.58269 | C>T | A>V | 97 | RdRp |
| | | 28688 | 0.57987 | T>C | L>L | 139 | Nucleocapsid |
| | | 1397 | 0.53043 | G>A | V>I | 198 | NSP2 |
| April-2020 | 441 | 11083 | 0.79874 | G>T | L>F | 37 | NSP6 |
| | | 28311 | 0.71328 | C>T | P>L | 13 | Nucleocapsid |
| | | 3037 | 0.70595 | C>T | F>F | 106 | NSP3 |
| | | 23403 | 0.69774 | A>G | D>G | 614 | Spike |
| | | 14408 | 0.6971 | C>T | P>L | 323 | RdRp |
| | | 6312 | 0.6678 | C>A | T>K | 1198 | NSP3 |
| | | 13730 | 0.66587 | C>T | A>V | 97 | RdRp |
| | | 23929 | 0.65279 | C>T | Y>Y | 789 | Spike |
| | | 28881 | 0.53127 | G>A | R>K | 203 | Nucleocapsid |
| | | 28882 | 0.53127 | G>A | R>R | 203 | Nucleocapsid |
| May-2020 | 977 | 28881 | 0.66198 | G>A | R>K | 203 | Nucleocapsid |
| | | 28882 | 0.66198 | G>A | R>R | 203 | Nucleocapsid |
| | | 28883 | 0.66198 | G>C | G>R | 204 | Nucleocapsid |
| | | 25563 | 0.64183 | G>T | Q>H | 57 | ORF3a |
| | | 26735 | 0.56685 | C>T | Y>Y | 71 | Membrane |
| | | 18877 | 0.5533 | C>T | L>L | 280 | Exon |
| | | 313 | 0.54277 | C>T | L>L | 16 | NSP1 |
| | | 14408 | 0.54115 | C>T | P>L | 323 | RdRp |
| | | 5700 | 0.50567 | C>A | A>D | 994 | NSP3 |
| | | 13730 | 0.48254 | C>T | A>V | 97 | RdRp |
| June-2020 | 1062 | 28881 | 0.72623 | G>A | R>K | 203 | Nucleocapsid |
| | | 28883 | 0.71049 | G>C | G>R | 204 | Nucleocapsid |
| | | 28882 | 0.69816 | G>A | R>R | 203 | Nucleocapsid |
| | | 22444 | 0.67332 | C>T | D>D | 294 | Spike |
| | | 25563 | 0.67187 | G>T | Q>H | 57 | ORF3a |
| | | 18877 | 0.66299 | C>T | L>L | 280 | Exon |
| | | 26735 | 0.6606 | C>T | Y>Y | 71 | Membrane |
| | | 28854 | 0.6393 | C>T | S>L | 194 | Nucleocapsid |
| | | 313 | 0.54631 | C>T | L>L | 16 | NSP1 |
| | | 5700 | 0.53036 | C>A | A>D | 994 | NSP3 |
| July-2020 | 683 | 28881 | 0.86601 | G>A | R>K | 203 | Nucleocapsid |
| | | 28882 | 0.85618 | G>A | R>R | 203 | Nucleocapsid |
| | | 28883 | 0.85615 | G>C | G>R | 204 | Nucleocapsid |
| | | 25563 | 0.69252 | G>T | Q>H | 57 | ORF3a |
| | | 313 | 0.66456 | C>T | L>L | 16 | NSP1 |
| | | 18877 | 0.66359 | C>T | L>L | 280 | Exon |
| | | 5700 | 0.65981 | C>A | A>D | 994 | NSP3 |
| | | 26735 | 0.65467 | C>T | Y>Y | 71 | Membrane |
| | | 28854 | 0.61568 | C>T | S>L | 194 | Nucleocapsid |
| | | 22444 | 0.60236 | C>T | D>D | 294 | Spike |

(*Continued*)

**Table 1.** (Continued)

| Month | Number of Sequences | Genomic Coordinate | Entropy | Nucleotide Change | Amino Acid Change | Protein Coordinate | Coding Region |
|---|---|---|---|---|---|---|---|
| August-2020 | 632 | 28881 | 0.79095 | G>A | R>K | 203 | Nucleocapsid |
| | | 28883 | 0.78919 | G>C | G>R | 204 | Nucleocapsid |
| | | 28882 | 0.78061 | G>A | R>R | 203 | Nucleocapsid |
| | | 22444 | 0.62652 | C>T | D>D | 294 | Spike |
| | | 25563 | 0.62045 | G>T | Q>H | 57 | ORF3a |
| | | 28854 | 0.61586 | C>T | S>L | 194 | Nucleocapsid |
| | | 26735 | 0.61193 | C>T | Y>Y | 71 | Membrane |
| | | 313 | 0.6079 | C>T | L>L | 16 | NSP1 |
| | | 18877 | 0.6079 | C>T | L>L | 280 | Exon |
| | | 5700 | 0.60235 | C>A | A>D | 994 | NSP3 |
| September-2020 | 629 | 28881 | 0.7396 | G>A | R>K | 203 | Nucleocapsid |
| | | 28882 | 0.68911 | G>A | R>R | 203 | Nucleocapsid |
| | | 28883 | 0.67924 | G>C | G>R | 204 | Nucleocapsid |
| | | 25563 | 0.60326 | G>T | Q>H | 57 | ORF3a |
| | | 313 | 0.59785 | C>T | L>L | 16 | NSP1 |
| | | 5700 | 0.59193 | C>A | A>D | 994 | NSP3 |
| | | 22444 | 0.57955 | C>T | D>D | 294 | Spike |
| | | 28854 | 0.56792 | C>T | S>L | 194 | Nucleocapsid |
| | | 18877 | 0.56622 | C>T | L>L | 280 | Exon |
| | | 26735 | 0.56103 | C>T | Y>Y | 71 | Membrane |
| October-2020 | 380 | 28881 | 0.78752 | G>A | R>K | 203 | Nucleocapsid |
| | | 28882 | 0.70769 | G>A | R>R | 203 | Nucleocapsid |
| | | 28883 | 0.70769 | G>C | G>R | 204 | Nucleocapsid |
| | | 22444 | 0.64744 | C>T | D>D | 294 | Spike |
| | | 18877 | 0.6463 | C>T | L>L | 280 | Exon |
| | | 26735 | 0.6463 | C>T | Y>Y | 71 | Membrane |
| | | 25563 | 0.64465 | G>T | Q>H | 57 | ORF3a |
| | | 28854 | 0.64124 | C>T | S>L | 194 | Nucleocapsid |
| | | 8917 | 0.57761 | C>T | F>F | 121 | NSP4 |
| | | 9389 | 0.55503 | G>A | D>N | 279 | NSP4 |
| November-2020 | 452 | 22444 | 0.75515 | C>T | D>D | 294 | Spike |
| | | 28881 | 0.74527 | G>A | R>K | 203 | Nucleocapsid |
| | | 28854 | 0.69762 | C>T | S>L | 194 | Nucleocapsid |
| | | 18877 | 0.68886 | C>T | L>L | 280 | Exon |
| | | 26735 | 0.68657 | C>T | Y>Y | 71 | Membrane |
| | | 25563 | 0.68439 | G>T | Q>H | 57 | ORF3a |
| | | 1947 | 0.66982 | T>C | V>A | 381 | NSP2 |
| | | 28882 | 0.66551 | G>A | R>R | 203 | Nucleocapsid |
| | | 28883 | 0.66551 | G>C | G>R | 204 | Nucleocapsid |
| | | 3267 | 0.48539 | C>T | T>I | 183 | NSP3 |
| December-2020 | 983 | 28881 | 0.71656 | G>A | R>K | 203 | Nucleocapsid |
| | | 22444 | 0.71598 | C>T | D>D | 294 | Spike |
| | | 1947 | 0.71371 | T>C | V>A | 381 | NSP2 |
| | | 25563 | 0.68512 | G>T | Q>H | 57 | ORF3a |
| | | 18877 | 0.67905 | C>T | L>L | 280 | Exon |
| | | 26735 | 0.67871 | C>T | Y>Y | 71 | Membrane |
| | | 28854 | 0.67728 | C>T | S>L | 194 | Nucleocapsid |
| | | 28883 | 0.67009 | G>C | G>R | 204 | Nucleocapsid |
| | | 28882 | 0.65134 | G>A | R>R | 203 | Nucleocapsid |
| | | 26060 | 0.56206 | C>T | T>I | 223 | ORF3a |

(*Continued*)

**Table 1.** (Continued)

| Month | Number of Sequences | Genomic Coordinate | Entropy | Nucleotide Change | Amino Acid Change | Protein Coordinate | Coding Region |
|---|---|---|---|---|---|---|---|
| January-2021 | 500 | 28881 | 0.82738 | G>A | R>K | 203 | Nucleocapsid |
| | | 28882 | 0.71685 | G>A | R>R | 203 | Nucleocapsid |
| | | 18877 | 0.70613 | C>T | L>L | 280 | Exon |
| | | 25563 | 0.70613 | G>T | Q>H | 57 | ORF3a |
| | | 28883 | 0.70225 | G>C | G>R | 204 | Nucleocapsid |
| | | 22444 | 0.69315 | C>T | D>D | 294 | Spike |
| | | 26735 | 0.69315 | C>T | Y>Y | 71 | Membrane |
| | | 28854 | 0.69286 | C>T | S>L | 194 | Nucleocapsid |
| | | 3267 | 0.63605 | C>T | T>I | 183 | NSP3 |
| | | 21034 | 0.61845 | C>T | L>L | 126 | NSP16 |
| February-2021 | 980 | 28881 | 1.13342 | G>A, G>T | R>K, R>M | 203 | Nucleocapsid |
| | | 23604 | 1.02071 | C>A, C>G | P>H, P>R | 681 | Spike |
| | | 23012 | 0.82687 | G>C, G>A | E>Q, E>K | 484 | Spike |
| | | 24775 | 0.69608 | A>T, A>- | Q>H, Q>- | 1071 | Spike |
| | | 28882 | 0.68897 | G>A | R>R | 203 | Nucleocapsid |
| | | 28883 | 0.67724 | G>C | G>R | 204 | Nucleocapsid |
| | | 28280 | 0.66855 | G>T, G>C | D>Y, D>H | 3 | Nucleocapsid |
| | | 25469 | 0.65125 | C>T | S>L | 26 | ORF3a |
| | | 22444 | 0.6458 | C>T | D>D | 294 | Spike |
| | | 29402 | 0.64017 | G>T | D>Y | 377 | Nucleocapsid |
| March-2021 | 1907 | 28881 | 1.03262 | G>A, G>T | R>K, R>M | 203 | Nucleocapsid |
| | | 23604 | 1.01066 | C>A, C>G | P>H, P>R | 681 | Spike |
| | | 28280 | 0.91893 | G>T, G>C | D>Y, D>H | 3 | Nucleocapsid |
| | | 23012 | 0.88114 | G>C, G>A | E>Q, E>K | 484 | Spike |
| | | 26767 | 0.84724 | T>C, T>G | I>T, I>S | 82 | Membrane |
| | | 11296 | 0.82674 | T>G, T>- | F>L, F>- | 108 | NSP6 |
| | | 21987 | 0.80846 | G>A, G>- | G>D, G>- | 142 | Spike |
| | | 24775 | 0.80534 | A>T, A>- | Q>H, Q>- | 1071 | Spike |
| | | 25469 | 0.77293 | C>T | S>L | 26 | ORF3a |
| | | 22022 | 0.76572 | G>A | E>K | 154 | Spike |
| April-2021 | 3054 | 28253 | 1.13895 | C>A, C>T, C>- | F>L, F>F, F>- | 120 | ORF8 |
| | | 22034 | 0.89681 | A>G, A>- | R>G, R>- | 158 | Spike |
| | | 26767 | 0.89284 | T>C, T>G | I>T, I>S | 82 | Membrane |
| | | 21987 | 0.87431 | G>A, G>- | G>D, G>- | 142 | Spike |
| | | 28249 | 0.84388 | A>T, A>- | D>V, D>- | 119 | ORF8 |
| | | 24410 | 0.8167 | G>A | D>N | 950 | Spike |
| | | 22033 | 0.76607 | C>- | F>- | 157 | Spike |
| | | 22032 | 0.756 | T>- | F>- | 157 | Spike |
| | | 28248 | 0.71357 | G>- | D>- | 119 | ORF8 |
| | | 11418 | 0.70573 | T>C | V>A | 149 | NSP6 |
| May-2021 | 2408 | 28253 | 1.08851 | C>A, C>T, C>- | F>L, F>F, F>- | 120 | ORF8 |
| | | 22034 | 0.81429 | A>G, A>- | R>G, R>- | 158 | Spike |
| | | 28249 | 0.81342 | A>T, A>- | D>V, D>- | 119 | ORF8 |
| | | 21987 | 0.76579 | G>A | G>D | 142 | Spike |
| | | 11418 | 0.70413 | T>C | V>A | 149 | NSP6 |
| | | 9891 | 0.69625 | C>T | A>V | 446 | NSP4 |
| | | 22030 | 0.68573 | G>- | E>- | 156 | Spike |
| | | 28251 | 0.6755 | T>- | F>- | 120 | ORF8 |
| | | 5184 | 0.66981 | C>T | P>L | 822 | NSP3 |
| | | 11201 | 0.66818 | A>G | T>A | 77 | NSP6 |

*(Continued)*

**Table 1.** (Continued)

| Month | Number of Sequences | Genomic Coordinate | Entropy | Nucleotide Change | Amino Acid Change | Protein Coordinate | Coding Region |
|---|---|---|---|---|---|---|---|
| June-2021 | 1293 | 21987 | 1.0067 | G>A, G>- | G>D, G>- | 142 | Spike |
| | | 28253 | 0.98317 | C>A, C>- | F>L, F>- | 120 | ORF8 |
| | | 28249 | 0.81706 | A>T, A>- | D>V, D>- | 119 | ORF8 |
| | | 22034 | 0.81496 | A>G, A>- | R>G, R>- | 158 | Spike |
| | | 11418 | 0.70538 | T>C | V>A | 149 | NSP6 |
| | | 27874 | 0.70016 | C>T | T>I | 40 | ORF7b |
| | | 9891 | 0.69617 | C>T | A>V | 446 | NSP4 |
| | | 28916 | 0.69472 | G>T | G>C | 215 | Nucleocapsid |
| | | 11201 | 0.69311 | A>G | T>A | 77 | NSP6 |
| | | 9053 | 0.69268 | G>T | V>L | 167 | NSP4 |
| July-2021 | 632 | 21987 | 0.93091 | G>A, G>- | G>D, G>- | 142 | Spike |
| | | 28253 | 0.87833 | C>A, C>- | F>L, F>- | 120 | ORF8 |
| | | 28249 | 0.7564 | A>T, A>- | D>V, D>- | 119 | ORF8 |
| | | 28251 | 0.71349 | T>- | F>- | 120 | ORF8 |
| | | 28250 | 0.711 | T>- | D>- | 119 | ORF8 |
| | | 28252 | 0.70261 | T>- | F>- | 120 | ORF8 |
| | | 4181 | 0.68595 | G>T | A>S | 488 | NSP3 |
| | | 5184 | 0.68595 | C>T | P>L | 822 | NSP3 |
| | | 6402 | 0.68595 | C>T | P>L | 1228 | NSP3 |
| | | 7124 | 0.68595 | C>T | P>S | 1469 | NSP3 |
| August-2021 | 15 | 28253 | 0.70869 | C>A | F>L | 120 | ORF8 |
| | | 4181 | 0.69142 | G>T | A>S | 488 | NSP3 |
| | | 6402 | 0.69142 | C>T | P>L | 1228 | NSP3 |
| | | 7124 | 0.69142 | C>T | P>S | 1469 | NSP3 |
| | | 8986 | 0.69142 | C>T | D>D | 144 | NSP4 |
| | | 9053 | 0.69142 | G>T | V>L | 167 | NSP4 |
| | | 10029 | 0.69142 | C>T | T>I | 492 | NSP4 |
| | | 11201 | 0.69142 | A>G | T>A | 77 | NSP6 |
| | | 11332 | 0.69142 | A>G | V>V | 120 | NSP6 |
| | | 19220 | 0.69142 | C>T | A>V | 394 | Exon |
| September-2021 | 52 | 21846 | 0.69315 | C>T | T>I | 95 | Spike |
| | | 24410 | 0.68696 | G>A | D>N | 950 | Spike |
| | | 5184 | 0.60769 | C>T | P>L | 822 | NSP3 |
| | | 27874 | 0.59084 | C>T | T>I | 40 | ORF7b |
| | | 4181 | 0.57228 | G>T | A>S | 488 | NSP3 |
| | | 6402 | 0.57228 | C>T | P>L | 1228 | NSP3 |
| | | 7124 | 0.57228 | C>T | P>S | 1469 | NSP3 |
| | | 8986 | 0.57228 | C>T | D>D | 144 | NSP4 |
| | | 9053 | 0.57228 | G>T | V>L | 167 | NSP4 |
| | | 10029 | 0.57228 | C>T | T>I | 492 | NSP4 |

in S1 File respectively. The evolution of the virus genome in terms of entropy for both temporal and spatial analysis is another crucial result reported in this work. For example, from a temporal perspective E484Q/K which is a much circulating variant in India has evolved over time but is on the wane now while for spatial analysis it can be seen that E484Q is one of the most prevalent variant in West Bengal. These evolution are visualised in Figs 3 and 4 respectively. It is to be noted that due to the lack of appropriate number of sequences, temporal data of

**Table 2. List of top 10 hotspot mutations based on spatial analysis.**

| State | Number of Sequences | Genomic Coordinate | Entropy | Nucleotide Change | Amino Acid Change | Protein Coordinate | Coding Region |
|---|---|---|---|---|---|---|---|
| Maharashtra | 3674 | 28881 | 1.02173 | G>A, G>T | R>K, R>M | 203 | Nucleocapsid |
| | | 26767 | 0.92484 | T>C, T>G | I>T, I>S | 82 | Membrane |
| | | 23604 | 0.81242 | C>G | P>R | 681 | Spike |
| | | 28253 | 0.806 | C>- | F>- | 120 | ORF8 |
| | | 21987 | 0.79485 | G>A, G>- | G>D, G>- | 142 | Spike |
| | | 25469 | 0.7663 | C>T | S>L | 26 | ORF3a |
| | | 27638 | 0.70457 | T>C | V>A | 82 | ORF7a |
| | | 29402 | 0.70178 | G>T | D>Y | 377 | Nucleocapsid |
| | | 22917 | 0.69779 | T>G | L>R | 452 | Spike |
| | | 23012 | 0.67477 | G>C | E>Q | 484 | Spike |
| Telangana | 2506 | 28253 | 1.0594 | C>T, C>- | F>F, F>- | 120 | ORF8 |
| | | 28881 | 1.05196 | G>A, G>T | R>K, R>M | 203 | Nucleocapsid |
| | | 22034 | 0.92872 | A>G, A>- | R>G, R>- | 158 | Spike |
| | | 23604 | 0.83122 | C>G | P>R | 681 | Spike |
| | | 26767 | 0.74928 | T>C | I>T | 82 | Membrane |
| | | 24410 | 0.72581 | G>A | D>N | 950 | Spike |
| | | 29402 | 0.71226 | G>T | D>Y | 377 | Nucleocapsid |
| | | 22033 | 0.70621 | C>- | F>- | 157 | Spike |
| | | 27638 | 0.70183 | T>C | V>A | 82 | ORF7a |
| | | 22917 | 0.70114 | T>G | L>R | 452 | Spike |
| Gujarat | 2333 | 28881 | 0.98391 | G>A, G>T | R>K, R>M | 203 | Nucleocapsid |
| | | 28253 | 0.98023 | C>A, C>- | F>L, F>- | 120 | ORF8 |
| | | 23604 | 0.89132 | C>A, C>G | P>H, P>R | 681 | Spike |
| | | 26767 | 0.79834 | T>C | I>T | 82 | Membrane |
| | | 28249 | 0.78731 | A>- | D>- | 119 | ORF8 |
| | | 22034 | 0.76092 | A>- | R>- | 158 | Spike |
| | | 22033 | 0.74274 | C>- | F>- | 157 | Spike |
| | | 22032 | 0.74262 | T>- | F>- | 157 | Spike |
| | | 25469 | 0.71957 | C>T | S>L | 26 | ORF3a |
| | | 22029 | 0.71048 | A>- | E>- | 156 | Spike |
| West Bengal | 1637 | 28881 | 1.03445 | G>A, G>T | R>K, R>M | 203 | Nucleocapsid |
| | | 26767 | 0.99595 | T>G, T>C | I>T, I>S | 82 | Membrane |
| | | 23604 | 0.9359 | C>A, C>G | P>H, P>R | 681 | Spike |
| | | 28253 | 0.88971 | C>A, C>- | F>L, F>- | 120 | ORF8 |
| | | 21987 | 0.81006 | G>A, G>- | G>D, G>- | 142 | Spike |
| | | 22034 | 0.80702 | A>G, A>- | R>G, R>- | 158 | Spike |
| | | 28249 | 0.77084 | A>- | D>- | 119 | ORF8 |
| | | 22917 | 0.70438 | T>G | L>R | 452 | Spike |
| | | 29402 | 0.7006 | G>T | D>Y | 377 | Nucleocapsid |
| | | 27638 | 0.69709 | T>C | V>A | 82 | ORF7a |
| Delhi | 1240 | 28881 | 1.08218 | G>A, G>T | R>K, R>M | 203 | Nucleocapsid |
| | | 23604 | 0.94518 | C>A, C>G | P>H, P>R | 681 | Spike |
| | | 22444 | 0.76965 | C>T | D>D | 294 | Spike |
| | | 25563 | 0.76199 | G>T | Q>H | 57 | ORF3a |
| | | 26735 | 0.72004 | C>T | Y>Y | 71 | Membrane |
| | | 18877 | 0.71311 | C>T | L>L | 280 | Exon |
| | | 28854 | 0.70723 | C>T | S>L | 194 | Nucleocapsid |
| | | 1947 | 0.68719 | T>C | V>A | 381 | NSP2 |
| | | 26767 | 0.65229 | T>C | I>T | 82 | Membrane |
| | | 28883 | 0.63286 | G>C | G>R | 204 | Nucleocapsid |

*(Continued)*

**Table 2.** (Continued)

| State | Number of Sequences | Genomic Coordinate | Entropy | Nucleotide Change | Amino Acid Change | Protein Coordinate | Coding Region |
|---|---|---|---|---|---|---|---|
| Andhra Pradesh | 1077 | 28253 | 1.21902 | C>A, C>T, C>- | F>L, F>F, F>- | 120 | ORF8 |
| | | 22034 | 1.04209 | A>G, A>- | R>G, R>- | 158 | Spike |
| | | 28881 | 0.85363 | G>A, G>T | R>K, R>M | 203 | Nucleocapsid |
| | | 22033 | 0.78715 | C>- | F>- | 157 | Spike |
| | | 26767 | 0.73239 | T>C | I>T | 82 | Membrane |
| | | 23604 | 0.73117 | C>G | P>R | 681 | Spike |
| | | 28249 | 0.71674 | A>- | D>- | 119 | ORF8 |
| | | 22030 | 0.70822 | G>- | E>- | 156 | Spike |
| | | 22029 | 0.70261 | A>- | E>- | 156 | Spike |
| | | 22031 | 0.69313 | T>- | F>- | 157 | Spike |
| Karnataka | 520 | 28881 | 1.23964 | G>A, G>T | R>K, R>M | 203 | Nucleocapsid |
| | | 28253 | 0.98145 | C>A | F>L | 120 | ORF8 |
| | | 23604 | 0.8514 | C>G | P>R | 681 | Spike |
| | | 28882 | 0.81953 | G>A | R>R | 203 | Nucleocapsid |
| | | 28883 | 0.80388 | G>C | G>R | 204 | Nucleocapsid |
| | | 26767 | 0.70691 | T>C | I>T | 82 | Membrane |
| | | 28249 | 0.67368 | A>T, A>- | D>V, D>- | 119 | ORF8 |
| | | 29402 | 0.6736 | G>T | D>Y | 377 | Nucleocapsid |
| | | 22917 | 0.64897 | T>G | L>R | 452 | Spike |
| | | 25469 | 0.64897 | C>T | S>L | 26 | ORF3a |
| Rajasthan | 434 | 28881 | 0.99106 | G>A, G>T | R>K, R>M | 203 | Nucleocapsid |
| | | 28882 | 0.69671 | G>A | R>R | 203 | Nucleocapsid |
| | | 28883 | 0.68481 | G>C | G>R | 204 | Nucleocapsid |
| | | 22444 | 0.6518 | C>T | D>D | 294 | Spike |
| | | 25563 | 0.63888 | G>T | Q>H | 57 | ORF3a |
| | | 28854 | 0.61881 | C>T | S>L | 194 | Nucleocapsid |
| | | 26735 | 0.61318 | C>T | Y>Y | 71 | Membrane |
| | | 18877 | 0.61125 | C>T | L>L | 280 | Exon |
| | | 1947 | 0.59878 | T>C, T>- | V>A, V>- | 381 | NSP2 |
| | | 23604 | 0.53191 | C>G | P>R | 681 | Spike |
| TamilNadu | 423 | 28253 | 1.16453 | C>A, C>T | F>L, F>F | 120 | ORF8 |
| | | 28881 | 1.09273 | G>A, G>T | R>K, R>M | 203 | Nucleocapsid |
| | | 23604 | 0.88416 | C>A, C>G | P>H, P>R | 681 | Spike |
| | | 28461 | 0.875 | A>G | D>G | 63 | Nucleocapsid |
| | | 24410 | 0.85053 | G>A | D>N | 950 | Spike |
| | | 26767 | 0.75549 | T>C | I>T | 82 | Membrane |
| | | 21618 | 0.69881 | C>G | T>R | 19 | Spike |
| | | 15451 | 0.68935 | G>A | G>S | 671 | RdRp |
| | | 16466 | 0.68935 | C>T | P>L | 77 | Helicase |
| | | 29402 | 0.67288 | G>T | D>Y | 377 | Nucleocapsid |
| Punjab | 418 | 11296 | 1.06149 | T>G, T>- | F>L, F>- | 108 | NSP6 |
| | | 28095 | 0.89567 | A>T, A>- | K>*, K>- | 68 | ORF8 |
| | | 28881 | 0.77179 | G>A, G>T | R>K, R>M | 203 | Nucleocapsid |
| | | 28280 | 0.76015 | G>C | D>H | 3 | Nucleocapsid |
| | | 23604 | 0.75325 | C>A, C>G | P>H, P>R | 681 | Spike |
| | | 28281 | 0.74341 | A>T | D>V | 3 | Nucleocapsid |
| | | 11291 | 0.69623 | G>- | G>- | 107 | NSP6 |
| | | 11295 | 0.69059 | T>- | F>- | 108 | NSP6 |
| | | 21765 | 0.68075 | T>- | I>- | 68 | Spike |
| | | 11292 | 0.66789 | G>- | G>- | 107 | NSP6 |

*(Continued)*

**Table 2.** (Continued)

| State | Number of Sequences | Genomic Coordinate | Entropy | Nucleotide Change | Amino Acid Change | Protein Coordinate | Coding Region |
|---|---|---|---|---|---|---|---|
| Chhattisgarh | 364 | 28881 | 1.07226 | G>A, G>T | R>K, R>M | 203 | Nucleocapsid |
| | | 23604 | 0.94912 | C>A, C>G | P>H, P>R | 681 | Spike |
| | | 26767 | 0.91621 | T>C, T>G | I>T, I>S | 82 | Membrane |
| | | 24410 | 0.90113 | G>A, G>- | D>N, D>- | 950 | Spike |
| | | 28461 | 0.71677 | A>G | D>G | 63 | Nucleocapsid |
| | | 28253 | 0.70958 | C>- | F>- | 120 | ORF8 |
| | | 15451 | 0.706 | G>A | G>S | 671 | RdRp |
| | | 27638 | 0.70498 | T>C | V>A | 82 | ORF7a |
| | | 21618 | 0.70489 | C>G | T>R | 19 | Spike |
| | | 29402 | 0.70441 | G>T | D>Y | 377 | Nucleocapsid |
| Manipur | 270 | 28253 | 1.02447 | C>A, C>- | F>L, F>- | 120 | ORF8 |
| | | 21987 | 0.87608 | G>A | G>D | 142 | Spike |
| | | 21846 | 0.71297 | C>T | T>I | 95 | Spike |
| | | 28916 | 0.70747 | G>T | G>C | 215 | Nucleocapsid |
| | | 11201 | 0.69044 | A>G | T>A | 77 | NSP6 |
| | | 28250 | 0.69044 | T>- | D>- | 119 | ORF8 |
| | | 28251 | 0.69044 | T>- | F>- | 120 | ORF8 |
| | | 28252 | 0.69044 | T>- | F>- | 120 | ORF8 |
| | | 5184 | 0.68705 | C>T | P>L | 822 | NSP3 |
| | | 6402 | 0.68705 | C>T | P>L | 1228 | NSP3 |
| Odisha | 238 | 28881 | 1.15561 | G>A, G>T | R>K, R>M | 203 | Nucleocapsid |
| | | 28882 | 0.78669 | G>A | R>R | 203 | Nucleocapsid |
| | | 28883 | 0.78669 | G>C | G>R | 204 | Nucleocapsid |
| | | 23604 | 0.73028 | C>G | P>R | 681 | Spike |
| | | 29402 | 0.58678 | G>T | D>Y | 377 | Nucleocapsid |
| | | 8917 | 0.57992 | C>T | F>F | 121 | NSP4 |
| | | 26767 | 0.56936 | T>C | I>T | 82 | Membrane |
| | | 22917 | 0.56881 | T>G | L>R | 452 | Spike |
| | | 24410 | 0.56082 | G>A | D>N | 950 | Spike |
| | | 9389 | 0.55771 | G>A | D>N | 279 | NSP4 |
| Uttar Pradesh | 229 | 26767 | 1.15838 | T>C, T>- | I>T, I>- | 82 | Membrane |
| | | 21618 | 1.07939 | C>G, C>- | T>R, T>- | 19 | Spike |
| | | 27752 | 0.98545 | C>T, C>- | T>I, T>- | 120 | ORF7a |
| | | 27638 | 0.95253 | T>C, T>- | V>A, V>- | 82 | ORF7a |
| | | 21987 | 0.87393 | G>A | G>D | 142 | Spike |
| | | 21872 | 0.7677 | T>- | W>- | 104 | Spike |
| | | 27874 | 0.76694 | C>T | T>I | 40 | ORF7b |
| | | 11418 | 0.75432 | T>C | V>A | 149 | NSP6 |
| | | 9053 | 0.74627 | G>T | V>L | 167 | NSP4 |
| | | 28916 | 0.74627 | G>T | G>C | 215 | Nucleocapsid |
| Haryana | 193 | 28881 | 0.99908 | G>A, G>T | R>K, R>M | 203 | Nucleocapsid |
| | | 23604 | 0.82165 | C>A, C>G | P>H, P>R | 681 | Spike |
| | | 25563 | 0.71135 | G>T | Q>H | 57 | ORF3a |
| | | 22444 | 0.70452 | C>T | D>D | 294 | Spike |
| | | 18877 | 0.67876 | C>T | L>L | 280 | Exon |
| | | 26735 | 0.67876 | C>T | Y>Y | 71 | Membrane |
| | | 28854 | 0.67695 | C>T | S>L | 194 | Nucleocapsid |
| | | 1947 | 0.63651 | T>C | V>A | 381 | NSP2 |
| | | 28882 | 0.62134 | G>A | R>R | 203 | Nucleocapsid |
| | | 28883 | 0.62134 | G>C | G>R | 204 | Nucleocapsid |

(*Continued*)

**Table 2.** (Continued)

| State | Number of Sequences | Genomic Coordinate | Entropy | Nucleotide Change | Amino Acid Change | Protein Coordinate | Coding Region |
|---|---|---|---|---|---|---|---|
| Himachal Pradesh | 184 | 1947 | 1.00628 | T>C, T>- | V>A, V>- | 381 | NSP2 |
| | | 28881 | 0.8515 | G>A | R>K | 203 | Nucleocapsid |
| | | 22444 | 0.74302 | C>T | D>D | 294 | Spike |
| | | 28854 | 0.7196 | C>T | S>L | 194 | Nucleocapsid |
| | | 28882 | 0.69576 | G>A | R>R | 203 | Nucleocapsid |
| | | 28883 | 0.69576 | G>C | G>R | 204 | Nucleocapsid |
| | | 18877 | 0.68944 | C>T | L>L | 280 | Exon |
| | | 25563 | 0.68944 | G>T | Q>H | 57 | ORF3a |
| | | 26735 | 0.68735 | C>T | Y>Y | 71 | Membrane |
| | | 26060 | 0.62056 | C>T | T>I | 223 | ORF3a |
| Sikkim | 165 | 28253 | 1.05282 | C>A, C>- | F>L, F>- | 120 | ORF8 |
| | | 28249 | 0.85603 | A>- | D>- | 119 | ORF8 |
| | | 28881 | 0.82105 | G>T | R>M | 203 | Nucleocapsid |
| | | 21987 | 0.79807 | G>A | G>D | 142 | Spike |
| | | 23604 | 0.7316 | C>G | P>R | 681 | Spike |
| | | 28251 | 0.72301 | T>- | F>- | 120 | ORF8 |
| | | 28252 | 0.72301 | T>- | F>- | 120 | ORF8 |
| | | 26767 | 0.70343 | T>C | I>T | 82 | Membrane |
| | | 22034 | 0.69379 | A>- | R>- | 158 | Spike |
| | | 9891 | 0.6927 | C>T | A>V | 446 | NSP4 |
| Jammu and Kashmir | 164 | 28881 | 1.05025 | G>A, G>T | R>K, R>M | 203 | Nucleocapsid |
| | | 23604 | 1.02063 | C>A, C>G | P>H, P>R | 681 | Spike |
| | | 22444 | 0.81197 | C>T | D>D | 294 | Spike |
| | | 28280 | 0.79577 | G>C | D>H | 3 | Nucleocapsid |
| | | 11296 | 0.76392 | T>- | F>- | 108 | NSP6 |
| | | 21765 | 0.67275 | T>- | I>- | 68 | Spike |
| | | 18877 | 0.66944 | C>T | L>L | 280 | Exon |
| | | 25563 | 0.66944 | G>T | Q>H | 57 | ORF3a |
| | | 26735 | 0.66383 | C>T | Y>Y | 71 | Membrane |
| | | 28854 | 0.66079 | C>T | S>L | 194 | Nucleocapsid |
| Puducherry | 138 | 28253 | 0.97927 | C>A, C>T | F>L, F>F | 120 | ORF8 |
| | | 23604 | 0.76675 | C>G | P>R | 681 | Spike |
| | | 28881 | 0.74111 | G>A, G>T | R>K, R>M | 203 | Nucleocapsid |
| | | 21987 | 0.69501 | G>A | G>D | 142 | Spike |
| | | 15451 | 0.6866 | G>A | G>S | 671 | RdRp |
| | | 16466 | 0.6866 | C>T | P>L | 77 | Helicase |
| | | 5184 | 0.68486 | C>T | P>L | 822 | NSP3 |
| | | 28249 | 0.68291 | A>T | D>- | 119 | ORF8 |
| | | 26767 | 0.6806 | T>C | I>T | 82 | Membrane |
| | | 1191 | 0.62794 | C>T | P>L | 129 | NSP2 |
| Meghalaya | 135 | 28253 | 0.99245 | C>A, C>- | F>L, F>- | 120 | ORF8 |
| | | 21987 | 0.8842 | G>A | G>D | 142 | Spike |
| | | 28249 | 0.84253 | A>T, A>- | D>V, D>- | 119 | ORF8 |
| | | 22034 | 0.78249 | A>- | R>- | 158 | Spike |
| | | 9891 | 0.68543 | C>T | A>V | 446 | NSP4 |
| | | 11418 | 0.68543 | T>C | V>A | 149 | NSP6 |
| | | 5184 | 0.6736 | C>T | P>L | 822 | NSP3 |
| | | 26767 | 0.66499 | T>C | I>T | 82 | Membrane |
| | | 28250 | 0.66015 | T>- | D>- | 119 | ORF8 |
| | | 28251 | 0.66015 | T>- | F>- | 120 | ORF8 |

*(Continued)*

**Table 2.** (Continued)

| State | Number of Sequences | Genomic Coordinate | Entropy | Nucleotide Change | Amino Acid Change | Protein Coordinate | Coding Region |
|-------|---------------------|--------------------|---------|-------------------|-------------------|--------------------|----------------|
| Uttarakhand | 126 | 28881 | 1.03137 | G>A, G>T | R>K, R>M | 203 | Nucleocapsid |
| | | 1947 | 0.77067 | T>C | V>A | 381 | NSP2 |
| | | 23604 | 0.76724 | C>G | P>R | 681 | Spike |
| | | 22444 | 0.73219 | C>T | D>D | 294 | Spike |
| | | 25563 | 0.66976 | G>T | Q>H | 57 | ORF3a |
| | | 18877 | 0.62109 | C>T | L>L | 280 | Exon |
| | | 26735 | 0.62109 | C>T | Y>Y | 71 | Membrane |
| | | 28882 | 0.62109 | G>A | R>R | 203 | Nucleocapsid |
| | | 28883 | 0.62109 | G>C | G>R | 204 | Nucleocapsid |
| | | 28854 | 0.61478 | C>T | S>L | 194 | Nucleocapsid |
| Kerala | 106 | 28881 | 0.80484 | G>A | R>K | 203 | Nucleocapsid |
| | | 3037 | 0.69298 | C>T | F>F | 106 | NSP3 |
| | | 14408 | 0.69298 | C>T | P>L | 323 | RdRp |
| | | 23403 | 0.69298 | A>G | D>G | 614 | Spike |
| | | 11083 | 0.6759 | G>T | L>F | 37 | NSP6 |
| | | 1397 | 0.6299 | G>A | V>I | 198 | NSP2 |
| | | 8653 | 0.6299 | G>T | M>I | 33 | NSP4 |
| | | 28688 | 0.6229 | T>C | L>L | 139 | Nucleocapsid |
| | | 884 | 0.6155 | C>T | R>C | 27 | NSP2 |
| | | 28883 | 0.59118 | G>C | G>R | 204 | Nucleocapsid |
| Madya Pradesh | 109 | 28881 | 0.98373 | G>A, G>T | R>K, R>M | 203 | Nucleocapsid |
| | | 23604 | 0.8576 | C>A, C>G | P>H, P>R | 681 | Spike |
| | | 28280 | 0.54646 | G>C | D>H | 3 | Nucleocapsid |
| | | 28882 | 0.52208 | G>A | R>R | 203 | Nucleocapsid |
| | | 28883 | 0.52208 | G>C | G>R | 204 | Nucleocapsid |
| | | 21895 | 0.51534 | T>C | D>D | 111 | Spike |
| | | 22917 | 0.51023 | T>G | L>R | 452 | Spike |
| | | 25469 | 0.51023 | C>T | S>L | 26 | ORF3a |
| | | 27638 | 0.51023 | T>C | V>A | 82 | ORF7a |
| | | 29402 | 0.51023 | G>T | D>Y | 377 | Nucleocapsid |
| Chandigarh | 102 | 22444 | 0.76942 | C>T | D>D | 294 | Spike |
| | | 11296 | 0.75797 | T>G | F>L | 108 | NSP6 |
| | | 28881 | 0.7328 | G>A | R>K | 203 | Nucleocapsid |
| | | 28882 | 0.68648 | G>A | R>R | 203 | Nucleocapsid |
| | | 28883 | 0.68648 | G>C | G>R | 204 | Nucleocapsid |
| | | 11291 | 0.68145 | G>- | G>- | 107 | NSP6 |
| | | 26735 | 0.65645 | C>T | Y>Y | 71 | Membrane |
| | | 18877 | 0.65095 | C>T | L>L | 280 | Exon |
| | | 25563 | 0.65095 | G>T | Q>H | 57 | ORF3a |
| | | 28854 | 0.63871 | C>T | S>L | 194 | Nucleocapsid |
| Assam | 101 | 28253 | 1.0999 | C>A, C>- | F>L, F>- | 120 | ORF8 |
| | | 28249 | 1.05588 | A>T, A>- | D>V, D>- | 119 | ORF8 |
| | | 28881 | 0.96252 | G>A, G>T | R>K, R>M | 203 | Nucleocapsid |
| | | 21987 | 0.9478 | G>A | G>D | 142 | Spike |
| | | 26767 | 0.91189 | T>C | I>T | 82 | Membrane |
| | | 22034 | 0.78077 | A>- | R>- | 158 | Spike |
| | | 24410 | 0.77309 | G>A | D>N | 950 | Spike |
| | | 23604 | 0.76149 | C>G | P>R | 681 | Spike |
| | | 15451 | 0.73936 | G>A | G>S | 671 | RdRp |
| | | 21618 | 0.73936 | C>G | T>R | 19 | Spike |

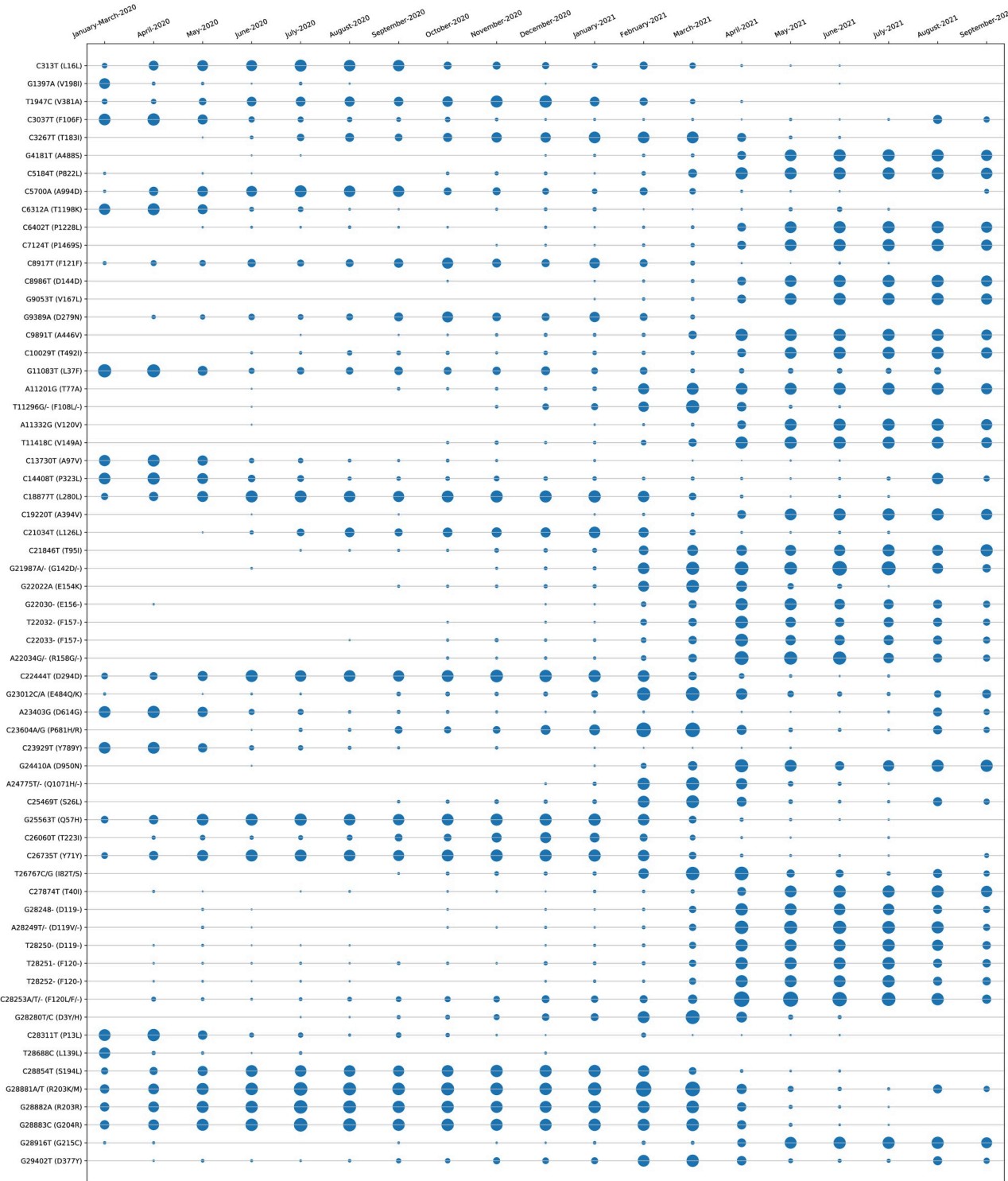

**Fig 3. Month wise (temporal) entropy of Indian SARS-CoV-2 genomes to show the changes in non-synonymous hotspot mutations.**

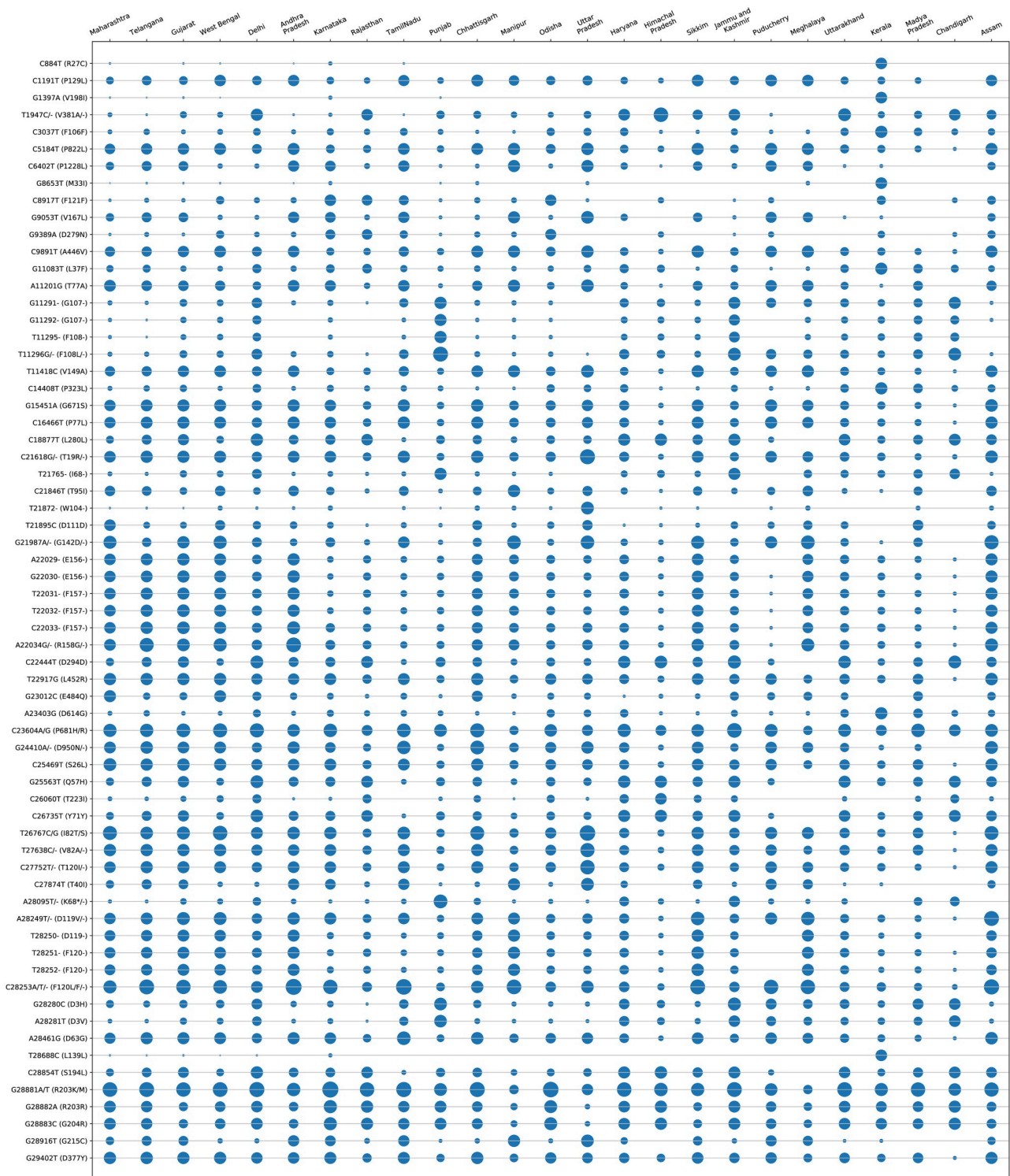

**Fig 4. State wise (spatial) entropy of Indian SARS-CoV-2 genomes to show the changes in non-synonymous hotspot mutations.**

January to March 2020 have been merged for the analysis. Please also note that non-coding regions of SARS-CoV-2 do not produce any protein to bind with human proteins. Thus, they are not considered for hotpot mutations. Moreover, since entropy calculation is performed on aligned sequences, only coding regions are considered for identification of hotspot mutations as the non-coding regions exhibit high entropy values and can be misleading while selecting such mutation points as hotspot mutations.

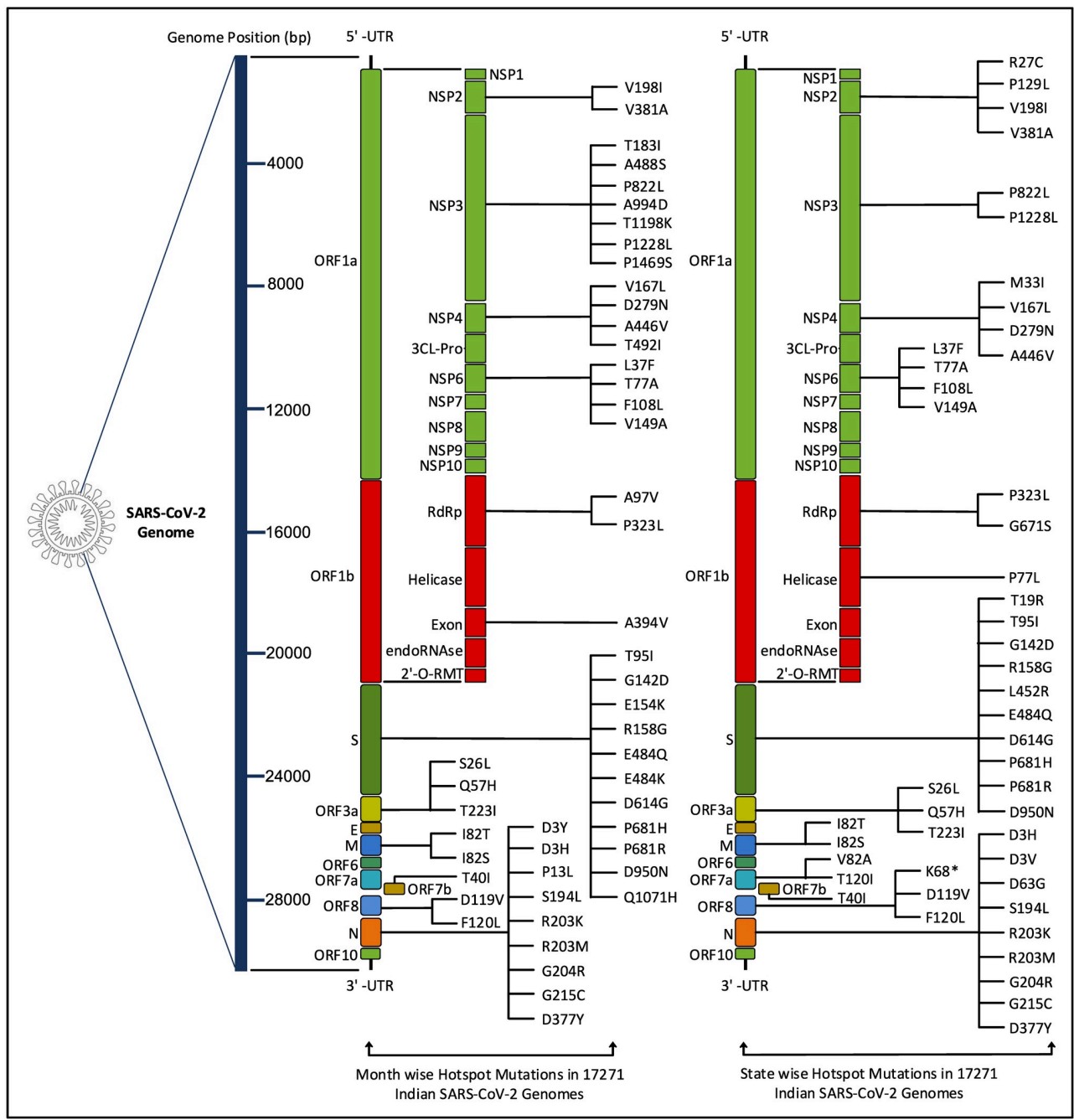

**Fig 5. Illustration of amino acid changes in SARS-CoV-2 proteins for the temporal and spatial non-synonymous hotspot mutations.**

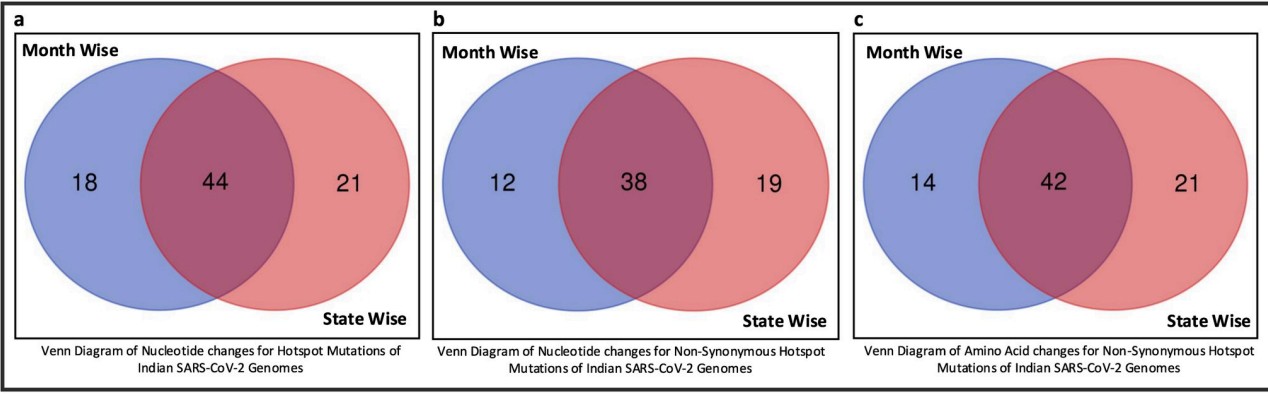

**Fig 6.** Venn diagrams of Indian SARS-CoV-2 Genomes to represent common (a) Nucleotide (b) Non-synonymous mutations and (c) Amino acid changes for the hotspot mutations.

Once the top 10 temporal and spatial hotspot mutations are identified, thereafter, 62 and 65 unique hotspot mutations are identified respectively for each category from 190 and 250 mutation points. For temporal analysis, 62 unique mutations result in 50 non-synonymous deletions and substitutions with corresponding 8 and 48 amino acid changes while for spatial analysis 57 non-synonymous deletions and substitutions are identified from 65 unique mutations with corresponding 16 and 47 amino acid changes. These non-synonymous mutations along with their amino acid changes in protein are visualised in Fig 5. Fig 6(a) depicts the common and unique nucleotide changes for all hotspot mutations for temporal and spatial analysis in the form of Venn diagram while Fig 6(b) shows the common and unique nucleotide changes for non-synonymous hotspot mutations and the common and unique amino acid changes in protein for such analysis are visualised in Fig 6(c). Fig 6(a) shows that there are 18 and 21 unique hotspot mutations considering temporal and spatial analysis while the number of such common mutations are 44. Fig 6(b) depicts 12 and 19 unique non-synonymous hotspot mutations while 38 changes are common in both. Finally, Fig 6(c) shows that there are unique 14 and 21 amino acid changes for temporal and spatial analysis with 42 changes common in both. All the amino acid changes in the protein for the non-synonymous hotspot mutations for temporal analysis are highlighted in Fig 7 while such mutations for the spatial analysis are shown in Fig 8. Please note that though 48 and 47 substitutions corresponding to temporal and spatial analysis are reported in Figs 5 and 6, only 47 and 46 such changes are highlighted in Figs 7 and 8 respectively. This is because the structure for ORF7b is not found in the literature and thus the corresponding hotspot mutation in the structure of ORF7b cannot be highlighted in either of the cases.

## Discussion

India has gone through the second wave of the SARS-CoV-2 pandemic and according to experts a third wave is inevitable as the virus is evolving and new strains are being identified. Thus, the study of the evolving virus strains is very crucial in the current pandemic scenario, In this regard, we have performed temporal and spatial analysis of 17271 SARS-CoV-2 sequences which has resulted in the identification of hotspot mutation points as SNPs in each category.

Changes in protein translations which can lead to functional instability in proteins are often attributed to structural alterations in amino acid residues. In this regard, to judge the

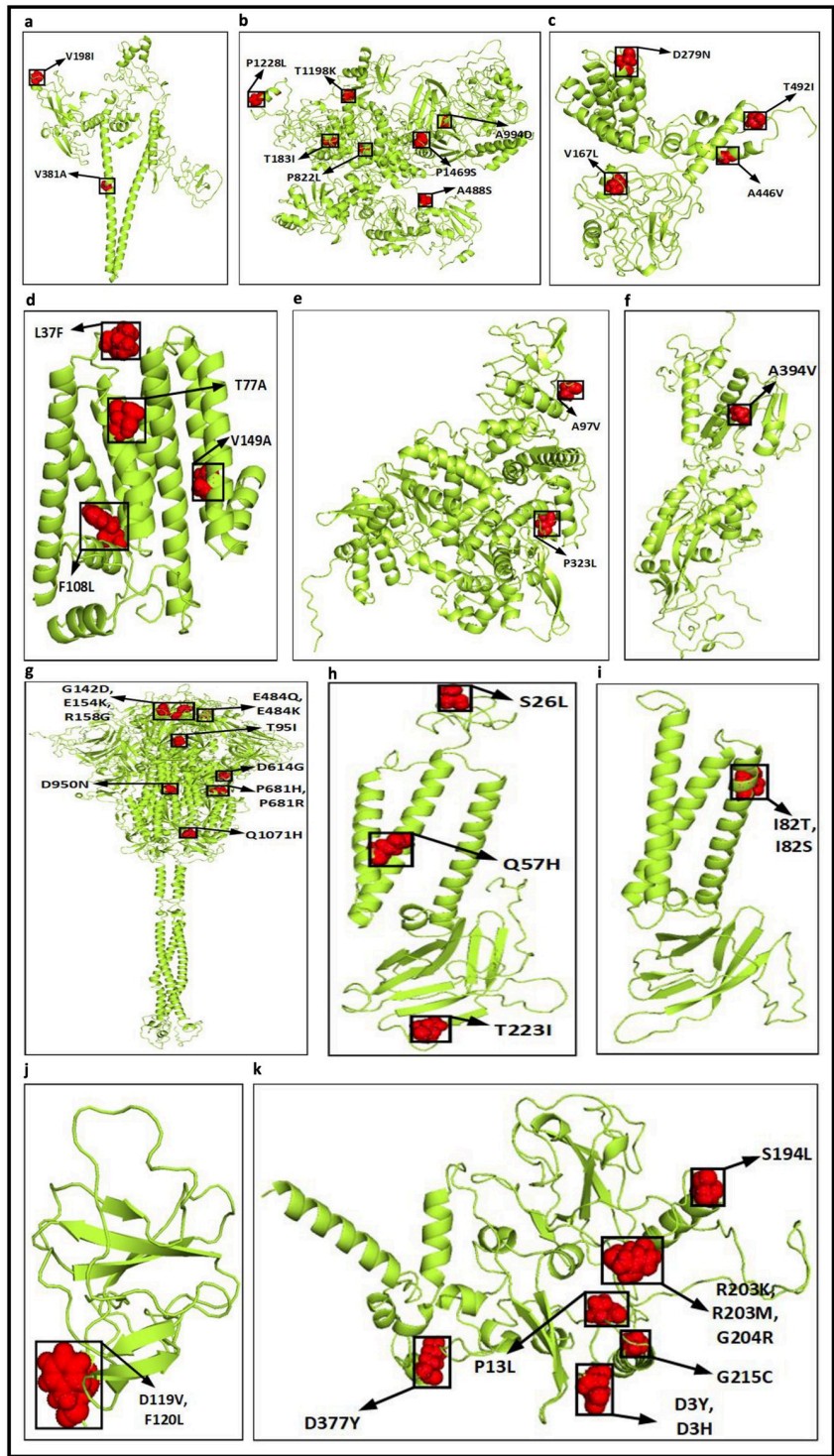

**Fig 7.** Highlighted amino acid changes in the protein structures for the non-synonymous hotspot mutations based on temporal analysis for (a) NSP2 (b) NSP3 (c) NSP4 (d) NSP6 (e) RdRp (f) Exon (g) Spike (h) ORF3a (i) Membrane (j) ORF8 (k) Nucleocapsid.

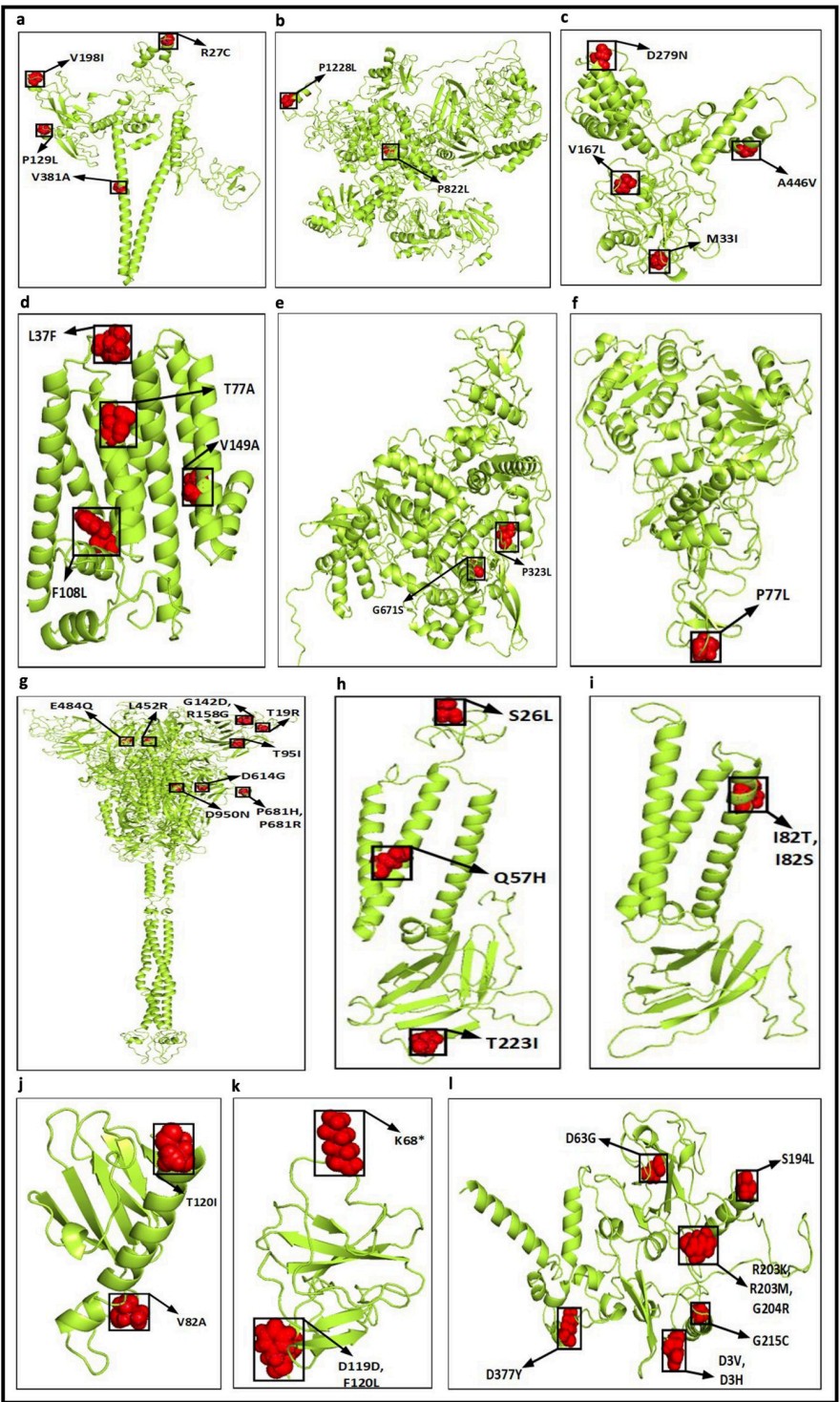

**Fig 8.** Highlighted amino acid changes in the protein structures for the non-synonymous hotspot mutations based on spatial analysis for (a) NSP2 (b) NSP3 (c) NSP4 (d) NSP6 (e) RdRp (f) Helicase (g) Spike (h) ORF3a (i) Membrane (j) ORF7a (k) ORF8 (l) Nucleocapsid.

**Table 3. Characteristics of non-synonymous hotspot mutations for temporal analysis.**

| Change in Nucleotide | Change in Amino Acid | Mapped with Coding Regions | PolyPhen-2 | | I-Mutant 2.0 | |
|---|---|---|---|---|---|---|
| | | | Prediction | Score | Stability | DDG |
| G1397A | V198I | NSP2 | Benign | 0.006 | Increase | 0.18 |
| T1947C | V381A | NSP2 | Benign | 0.009 | Decrease | -1.64 |
| C3267T | T183I | NSP3 | NG | NG | Decrease | -0.1 |
| G4181T | A488S | NSP3 | Benign | 0.017 | Decrease | -0.89 |
| C5184T | P822L | NSP3 | Benign | 0.011 | Decrease | -0.54 |
| C5700A | A994D | NSP3 | Possibly Damaging | 0.935 | Decrease | -0.78 |
| C6312A | T1198K | NSP3 | Probably Damaging | 0.998 | Decrease | -1.37 |
| C6402T | P1228L | NSP3 | Benign | 0.001 | Decrease | -0.46 |
| C7124T | P1469S | NSP3 | Probably Damaging | 0.967 | Decrease | -2.17 |
| G9053T | V167L | NSP4 | Benign | 0.406 | Decrease | -2.14 |
| G9389A | D279N | NSP4 | Probably Damaging | 0.999 | Decrease | -1.26 |
| C9891T | A446V | NSP4 | Probably Damaging | 0.999 | Increase | 0.64 |
| C10029T | T492I | NSP4 | Probably Damaging | 0.973 | Decrease | -0.08 |
| G11083T | L37F | NSP6 | Benign | 0.027 | Decrease | -0.05 |
| A11201G | T77A | NSP6 | Possibly Damaging | 0.577 | Decrease | -0.7 |
| T11296G | F108L | NSP6 | Benign | 0.001 | Decrease | -3.31 |
| T11418C | V149A | NSP6 | Possibly Damaging | 0.865 | Decrease | -3.43 |
| C13730T | A97V | RdRp | Probably Damaging | 0.99 | Decrease | -0.53 |
| C14408T | P323L | RdRp | Benign | 0.018 | Decrease | -0.8 |
| C19220T | A394V | Exon | Benign | 0.005 | Decrease | -0.17 |
| C21846T | T95I | Spike | Probably Damaging | 0.999 | Decrease | -1.8 |
| G21987A | G142D | Spike | Benign | 0.061 | Decrease | -1.17 |
| G22022A | E154K | Spike | NG | NG | Decrease | -1.4 |
| A22034G | R158G | Spike | NG | NG | Decrease | -2.63 |
| G23012C | E484Q | Spike | Possibly Damaging | 0.881 | Decrease | -0.48 |
| G23012A | E484K | Spike | Possibly Damaging | 0.601 | Decrease | -0.85 |
| A23403G | D614G | Spike | Benign | 0.004 | Decrease | -1.94 |
| C23604A | P681H | Spike | NG | NG | Decrease | -0.92 |
| C23604G | P681R | Spike | NG | NG | Decrease | -0.79 |
| G24410A | D950N | Spike | Benign | 0.34 | Increase | 0.15 |
| A24775T | Q1071H | Spike | Probably Damaging | 0.997 | Decrease | -1.19 |
| C25469T | S26L | ORF3a | Benign | 0.017 | Increase | 0.92 |
| G25563T | Q57H | ORF3a | Probably Damaging | 0.983 | Decrease | -1.12 |
| C26060T | T223I | ORF3a | Probably Damaging | 0.998 | Decrease | -0.07 |
| T26767G | I82S | Membrane | Possibly Damaging | 0.951 | Decrease | -2 |
| T26767C | I82T | Membrane | Possibly Damaging | 0.889 | Decrease | -2.41 |
| C27874T | T40I | ORF7b | NG | NG | Decrease | -0.22 |
| A28249T | D119V | ORF8 | Possibly Damaging | 0.541 | Decrease | -0.63 |
| C28253A | F120L | ORF8 | Probably Damaging | 0.988 | Decrease | -2.95 |
| G28280T | D3Y | Nucleocapsid | Probably Damaging | 1 | Increase | 0.22 |
| G28280C | D3H | Nucleocapsid | Probably Damaging | 1 | Increase | 0.34 |
| C28311T | P13L | Nucleocapsid | Probably Damaging | 1 | Increase | 0.11 |
| C28854T | S194L | Nucleocapsid | Probably Damaging | 0.994 | Increase | 0.45 |
| G28881A | R203K | Nucleocapsid | Probably Damaging | 0.969 | Decrease | -2.26 |
| G28881T | R203M | Nucleocapsid | Probably Damaging | 0.998 | Decrease | -1.52 |
| G28883C | G204R | Nucleocapsid | Probably Damaging | 1 | No Change | 0 |

*(Continued)*

**Table 3.** (Continued)

| Change in Nucleotide | Change in Amino Acid | Mapped with Coding Regions | PolyPhen-2 | | I-Mutant 2.0 | |
|---|---|---|---|---|---|---|
| | | | Prediction | Score | Stability | DDG |
| G28916T | G215C | Nucleocapsid | Probably Damaging | 1 | Decrease | -0.49 |
| G29402T | D377Y | Nucleocapsid | Probably Damaging | 1 | Increase | 0.51 |

functional characteristics of all the non-synonymous hotspot mutations, their changes in proteins are evaluated as biological functions considering the sequences by using PolyPhen-2 (Polymorphism Phenotyping) [21] while I-Mutant 2.0 [22] evaluates their structural stability. Such results for temporal and spatial analysis are reported in Tables 3 and 4 respectively. The tools used for such prediction are PolyPhen-2 and I-Mutant 2.0. The prediction of Polyphen-2 http://genetics.bwh.harvard.edu/pph2/ works with sequence, structural and phylogenetic information of a SNP while I-Mutant 2.0 https://folding.biofold.org/i-mutant/i-mutant2.0. html uses support vector machine (SVM) for the automatic prediction of protein stability changes upon single point mutations. PolyPhen-2 is used to find the damaging non-synonymous hotspot mutations while protein stabilities are determined by I-Mutant 2.0. The score generated by Polyphen-2 lies between the range of 0 to 1. A score close to 1 denotes that the mutations can be more confidently considered to be damaging. Considering the prediction of Polyphen-2, it can be seen from Table 3 that out of the 56 unique amino acid changes, 27 changes are damaging for temporal analysis while for spatial analysis as can be seen from Table 4, out of 63 unique amino acid changes, 24 changes are damaging. It is important to note that in case of protein, damaging mostly defines instability. Generally, this is used for human proteins. As a consequence, if the human protein is damaging in nature because of mutations, then the human protein-protein interactions may occur with high or low binding affinity. Now in case of virus, similar consequences may happen which means if the virus protein is damaged because of mutations, it may interact with human proteins with similar binding affinity. As a result, the virus may acquire characteristics like transmissibility, escaping antibodies [23, 24] etc.

Stability is yet another parameter which is crucial to judge the functional and structural activity of a protein. Protein stability dictates the conformational structure of the protein, thereby determining its function. Any change in protein stability may cause misfolding, degradation or aberrant conglomeration of proteins. In I-Mutant 2.0 the changes in the protein stability is predicted using free energy change values (DDG). A zero or a negative value of DDG indicates that the stability of a protein is decreasing. The result from I-mutant 2.0 infers that of the 27 and 24 unique deleterious or damaging changes for temporal and spatial analysis, 21 changes for both decrease the stability of the protein structures. The common mutations in both the categories are T77A and V149A in NSP6, T95I and E484Q in Spike, Q57H and T223I in ORF3a, I82S and I82T in Membrane, D119V and F120L in ORF8, R203K, R203M and G215C in Nucleocapsid. It is to be noted that, apart from these mutations, other important mutations as recognised by virologists in the multiple variants of concern like Alpha, Beta and Delta are L452R, E484K, D614G, P681H and P681R in Spike.

Furthermore, the entropy change of the hotspot mutations for the different variants like Alpha, Beta and Delta are shown in Fig 9(a)–9(c) respectively. For example, hotspot mutation E484K in Alpha variant in Fig 9(a) which was dominant in the months of February-April 2021 has declined over the next few months. Also, D614G which is a common hotspot mutation in all the variants has also declined over time. Moreover, mutations like L452R and P681R which are part of the Delta variant are also two of the hotspot mutations as identified by the analysis.

**Table 4. Characteristics of non-synonymous hotspot mutations for spatial analysis.**

| Change in Nucleotide | Change in Amino Acid | Mapped with Coding Regions | PolyPhen-2 | | I-Mutant 2.0 | |
|---|---|---|---|---|---|---|
| | | | Prediction | Score | Stability | DDG |
| C884T | R27C | NSP2 | Probably Damaging | 1 | Decrease | -0.35 |
| C1191T | P129L | NSP2 | Possibly Damaging | 0.924 | Decrease | -0.53 |
| G1397A | V198I | NSP2 | Benign | 0.006 | Increase | 0.18 |
| T1947C | V381A | NSP2 | Benign | 0.009 | Decrease | -1.64 |
| C5184T | P822L | NSP3 | Benign | 0.011 | Decrease | -0.54 |
| C6402T | P1228L | NSP3 | Benign | 0.001 | Decrease | -0.46 |
| G8653T | M33I | NSP4 | Benign | 0.002 | Decrease | -0.73 |
| G9053T | V167L | NSP4 | Benign | 0.406 | Decrease | -2.14 |
| G9389A | D279N | NSP4 | Probably Damaging | 0.999 | Decrease | -1.26 |
| C9891T | A446V | NSP4 | Probably Damaging | 0.999 | Increase | 0.64 |
| G11083T | L37F | NSP6 | Benign | 0.027 | Decrease | -0.05 |
| A11201G | T77A | NSP6 | Possibly Damaging | 0.577 | Decrease | -0.7 |
| T11296G | F108L | NSP6 | Benign | 0.001 | Decrease | -3.31 |
| T11418C | V149A | NSP6 | Possibly Damaging | 0.865 | Decrease | -3.43 |
| C14408T | P323L | RdRp | Benign | 0.018 | Decrease | -0.8 |
| G15451A | G671S | RdRp | Probably Damaging | 1 | Decrease | -0.29 |
| C16466T | P77L | Helicase | Probably Damaging | 1 | Decrease | -1.03 |
| C21618G | T19R | Spike | Benign | 0.007 | Decrease | -0.12 |
| C21846T | T95I | Spike | Probably Damaging | 0.999 | Decrease | -1.8 |
| G21987A | G142D | Spike | Benign | 0.061 | Decrease | -1.17 |
| A22034G | R158G | Spike | NG | NG | Decrease | -2.63 |
| T22917G | L452R | Spike | Benign | 0.017 | Decrease | -1.4 |
| G23012C | E484Q | Spike | Possibly Damaging | 0.881 | Decrease | -0.48 |
| A23403G | D614G | Spike | Benign | 0.004 | Decrease | -1.94 |
| C23604A | P681H | Spike | NG | NG | Decrease | -0.92 |
| C23604G | P681R | Spike | NG | NG | Decrease | -0.79 |
| G24410A | D950N | Spike | Benign | 0.34 | Increase | 0.15 |
| C25469T | S26L | ORF3a | Benign | 0.017 | Increase | 0.92 |
| G25563T | Q57H | ORF3a | Probably Damaging | 0.983 | Decrease | -1.12 |
| C26060T | T223I | ORF3a | Probably Damaging | 0.998 | Decrease | -0.07 |
| T26767G | I82S | Membrane | Possibly Damaging | 0.951 | Decrease | -2 |
| T26767C | I82T | Membrane | Possibly Damaging | 0.889 | Decrease | -2.41 |
| T27638C | V82A | ORF7a | Possibly Damaging | 0.732 | Decrease | -2.18 |
| C27752T | T120I | ORF7a | Possibly Damaging | 0.915 | Decrease | -0.26 |
| C27874T | T40I | ORF7b | NG | NG | Decrease | -0.22 |
| A28249T | D119V | ORF8 | Possibly Damaging | 0.541 | Decrease | -0.63 |
| C28253A | F120L | ORF8 | Probably Damaging | 0.988 | Decrease | -2.95 |
| G28280C | D3H | Nucleocapsid | Probably Damaging | 1 | Increase | 0.34 |
| A28281T | D3V | Nucleocapsid | Probably Damaging | 1 | Decrease | -0.22 |
| A28461G | D63G | Nucleocapsid | Benign | 0 | Decrease | -0.57 |
| C28854T | S194L | Nucleocapsid | Probably Damaging | 0.994 | Increase | 0.45 |
| G28881A | R203K | Nucleocapsid | Probably Damaging | 0.969 | Decrease | -2.26 |
| G28881T | R203M | Nucleocapsid | Probably Damaging | 0.998 | Decrease | -1.52 |
| G28883C | G204R | Nucleocapsid | Probably Damaging | 1 | No Change | 0 |
| G28916T | G215C | Nucleocapsid | Probably Damaging | 1 | Decrease | -0.49 |
| G29402T | D377Y | Nucleocapsid | Probably Damaging | 1 | Increase | 0.51 |

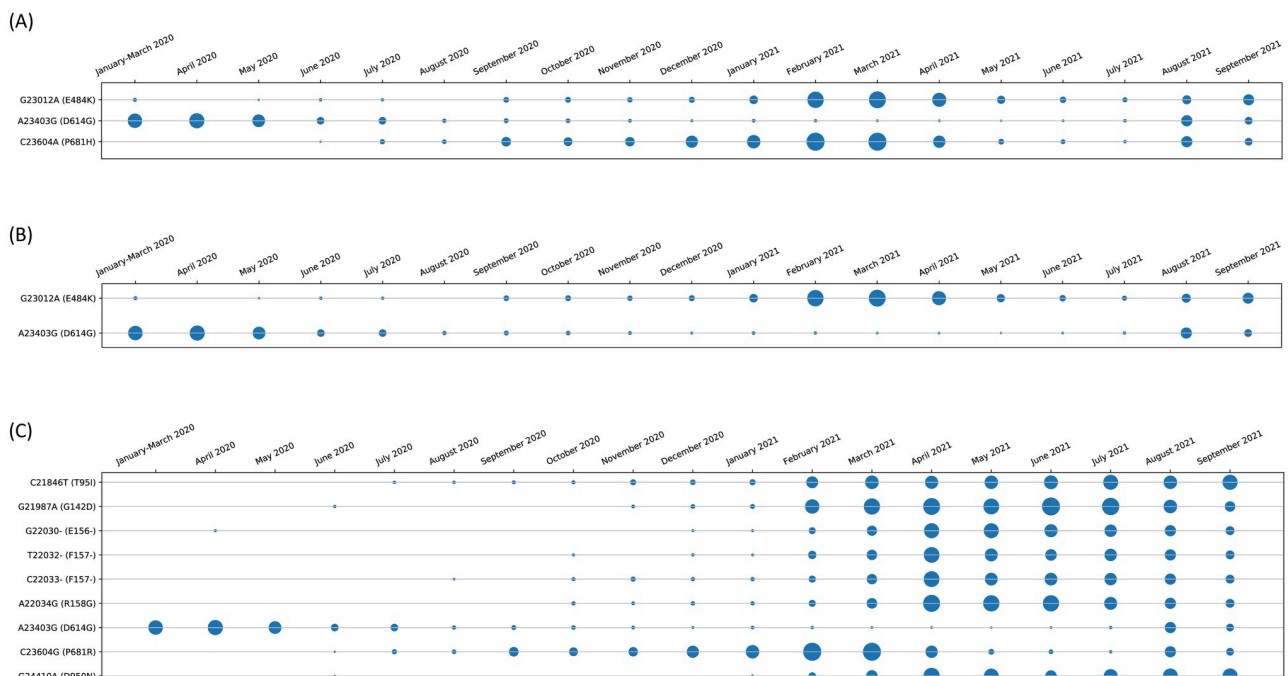

**Fig 9.** Month wise evolution of (a) Alpha (B.1.1.7) (b) Beta (B.1.351) and (c) Delta (B.1.617.2) variants for non-synonymous hotspot mutations.

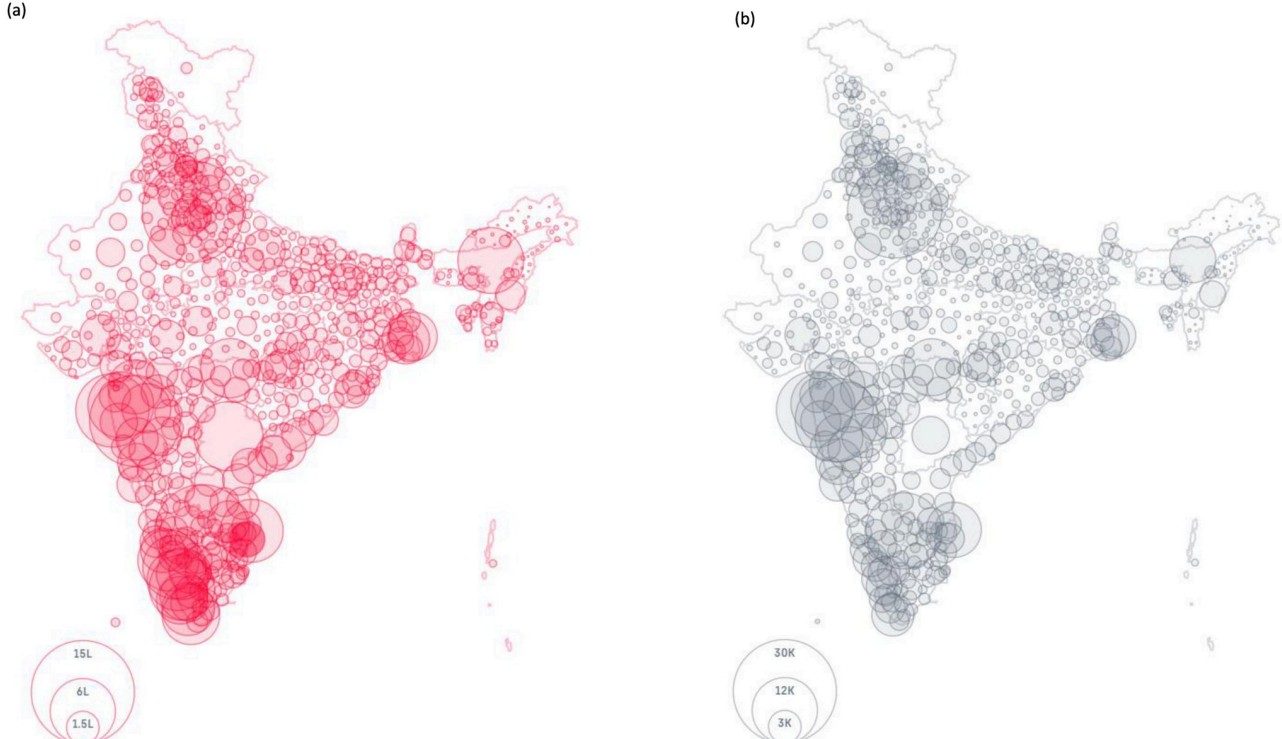

**Fig 10.** Illustration of (a) Confirmed and (b) Deceased cases of India to show the effects of SARS-CoV-2 in the different regions of the country.

It is to be noted that Delta variant was responsible for the catastrophic 2nd wave in India. Fig 10(a) and 10(b) show the plot of confirmed and deceased cases in India till 31st October 2021. For example, western part of India has a very high number of confirmed and deceased cases which can be attributed to the Delta variant. As is shown in Table 2, Maharashtra which lies in the western part of India has both of the aforementioned mutations identified as hotspots. All these figures are considered from https://www.covid19india.org/.

## Conclusion

As the second wave of COVID pandemic had hit India really hard, understanding the evolution of SARS-CoV-2 virus is most crucial in this scenario. In this regard, temporal (month-wise) and spatial (state-wise) analysis are carried out for 17271 aligned Indian sequences to identify top 10 hotspot mutation points in the coding regions based on entropy for each month as well as for each state. Additionally, to judge the functional characteristics of all the non-synonymous hotspot mutations, their changes in proteins are evaluated as biological functions considering the sequences by using PolyPhen-2 while I-Mutant 2.0 evaluates their structural stability. As a result, for both temporal and spatial analysis, the common damaging and unstable mutations are T77A and V149A in NSP6, T95I and E484Q in Spike, Q57H and T223I in ORF3a, I82S and I82T in Membrane, D119V and F120L in ORF8, R203K, R203M and G215C in Nucleocapsid. Also, investigation of the effects of the characteristics of the hotspot mutations of SARS-CoV-2 on human hosts can be conducted with the help of virologists. The authors are working in this direction as well.

## Supporting information

**S1 File. This file contains 4 supplementary tables named as S1-S4.**
(PDF)

## Acknowledgments

We thank all those who have contributed sequences to GISAID database.

## Author Contributions

**Conceptualization:** Nimisha Ghosh, Suman Nandi, Indrajit Saha.

**Data curation:** Nimisha Ghosh, Indrajit Saha.

**Formal analysis:** Nimisha Ghosh, Suman Nandi, Indrajit Saha.

**Funding acquisition:** Nimisha Ghosh, Indrajit Saha.

**Investigation:** Indrajit Saha.

**Methodology:** Nimisha Ghosh, Indrajit Saha.

**Project administration:** Indrajit Saha.

**Resources:** Indrajit Saha.

**Software:** Nimisha Ghosh, Suman Nandi.

**Supervision:** Indrajit Saha.

**Validation:** Nimisha Ghosh, Suman Nandi, Indrajit Saha.

**Visualization:** Suman Nandi, Indrajit Saha.

**Writing – original draft:** Nimisha Ghosh.

**Writing – review & editing:** Suman Nandi, Indrajit Saha.

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
