## [Decision Letter · Decision Letter 0]

27 Aug 2021

PONE-D-21-17506

Phylogenetic Analysis of 5734 Indian SARS-CoV-2 Genomes to Identify Temporal and Spatial Hotspot Mutations

PLOS ONE

Dear Dr. Ghosh,

Thank you for submitting your manuscript to PLOS ONE. After careful consideration, we feel that it has merit but does not fully meet PLOS ONE’s publication criteria as it currently stands. Therefore, we invite you to submit a revised version of the manuscript that addresses the points raised during the review process.

Although the manuscript is well-prepared and timely, some important concerns need to be addressed. The authors should account for writing the manuscript clearly and provide appropriate discussion.

We look forward to receiving your revised manuscript.

Kind regards,

Arunachalam Ramaiah, PhD

Academic Editor

PLOS ONE

Journal Requirements:

Reviewers' comments:

Reviewer's Responses to Questions

**Comments to the Author**

1. Is the manuscript technically sound, and do the data support the conclusions?

Reviewer #1: Yes

Reviewer #2: No

Reviewer #3: Partly

2. Has the statistical analysis been performed appropriately and rigorously? 

Reviewer #1: Yes

Reviewer #2: No

Reviewer #3: I Don't Know

3. Have the authors made all data underlying the findings in their manuscript fully available?

Reviewer #1: Yes

Reviewer #2: Yes

Reviewer #3: Yes

4. Is the manuscript presented in an intelligible fashion and written in standard English?

Reviewer #1: Yes

Reviewer #2: Yes

Reviewer #3: No

5. Review Comments to the Author

Reviewer #1: SARS-CoV-2 is still spreading around the world very rapidly with a high infection rate that could become a global pandemic. In order for researchers, including us, to design a more effective vaccine, we are analyzing the evolving virus strains.

You have performed multiple sequence alignments of 5734 sequences of SARS-CoV-2 using MAFFT and phylogenetic analysis using Nextstrain. Then, we identified several SNPs and point mutations. You identified the top 10 hotspot mutation points in the coding region. As a result, 130 hotspot mutations were identified in the temporal analysis and 250 in the spatial analysis. Subsequently, 32 temporally unique and 63 spatially unique hotspot mutations were identified, respectively. In addition, you have identified 21 point mutations in the time series analysis. For example, A97V in RdRp, L126F in NSP16, Q57H in ORF3a, and R203K, R203M, and G204R in Nucleocapsid. You also reaffirmed that the mutations of concern are E484Q and E484K in Spike.

minor issues

I could not read Tables 1 through 3 because the text was too small. Therefore, you should replace them with complete and readable tables in the text. Please move this table to the supplement.

I believe that your paper needs to be released to the world as soon as possible.

Reviewer #2: This paper analyzes GISAID data to identify mutational hotspots within SARS-CoV-2 genomes sequenced in India. This analysis identifies several mutations that appear to change over time or vary between regions within India. The authors claim that these locations are mutational hotspots but do not provide compelling evidence to support this.

Specific comments:

1. Analysis of mutational hotspots is confounded by the competition between variants. Specifically, when one variant displaces another in a state or over time this will appear to show an enrichment for mutations associated with this variant even if the location of these mutations is not functionally important. At least for the positions identified as mutation “hotspots”, the authors should test whether they are changing in frequency within specific lineages. As an example, S:E484Q within the Delta lineage was presumed to be particularly problematic since it resembles the S:E484K mutation found in the Beta variant but it has since declined in frequency within the Delta variant.

2. Please avoid the use of the term “double mutant” as it is scientifically misleading. Please refer to the strains by either their PANGO lineage (e.g. B.1.617.2) or by their WHO designation (e.g. Delta variant).

3. Figure captions need more description.

4. Variants refers to the combination of many mutations rather than any specific mutation. E.g. Page 2, line 93 is incorrect since E484Q is not a variant. Please fix terminology throughout.

5. The protein structure model shown for nucleocapsid indicates a mostly unstructured architecture, but this is misleading. The N-terminal and C-terminal domain structures have been solved by multiple research groups and are known to be well ordered. Please update with a revised structure.

6. The use of PROVEAN and similar tools to detect “damaging” mutations is not explained well and is potentially misleading. These tools were designed to detect the impact of mutations of human proteins rather than viral proteins with the assumption that major changes to human proteins are likely to be deleterious. This cannot be assumed for emerging viruses because they are under a rapidly changing selection conditions and mutations identified by PROVEAN might be beneficial for the virus to avoid immunity or even to enhance function. This needs to be discussed more clearly. It is also unclear how this helps to identify a location as a potential mutation hotspot.

Reviewer #3: Comments to the Author: The manuscript give a meaningful view point to the analysis of evolving virus strains of SARS-CoV-2, but the paper is not clearly written.

Major points:

The paper from introduction to discussion should be simplified; it’s too long and too verbose. The manuscript did not give a meaningful view about what authors do this work and what they get conclusion. I recommend author to revise the manuscript as clear as possible.

Abstract: I strongly recommend author to re-write the abstract and give a clear abstract.

Introduction: It’s too verbose. It present too much description to others research. The author may likely just put others researches together rather than summary and conclude their studies.

Line。。。： 300K should be replaced using a formal description, e.g 300,000. The problem is through the entire manuscript.

Line 10: The prevalent variant in South African is B.1.351, so “501Y. V2” should change to B.1.351.

Line 11: Japanese should be Japan. Brazilian should be Brazil.

Line 11: E484K is not a linage, please clarify update the mainly variants name.

Line 24: In [8] and In [9] should be replace as more reasonable description, It can be change by author name or change to another description. This problem is present through the whole paper.

Line 23-25: what this sentence relationship with previous viewpoint?

Line 26-31: The author would like to

Line 31-32: what this sentence mean “thereby indicating potential impacts on the ongoing development of various COVID-19 diagnosis and cure”? and what this sentence relationship with these variants?

Line 36-37: previous have mentioned that “they have found Nucleocapsid to have the highest mutational changes in frequency”, the author can summary them together rather than describe it again and again.

Line 15-72: All citation view were list, please summary and clarify the paragraph.

Line 75: the method “multiple alignment using fast fourier transform (MAFFT)” could move to method rather introduction.

Line 77-94: The sentence about method should be move to “Material and Methods” part, the sentence about the result details should be move to Results part. The introduction just retain the summary of resuts and meaning of this paper.

Line 105: The citation is a lab? And the reference list 4 was not contain an author named Zhang.

Line 115: “MAFFT which is a progressive alignment technique is used as the multiple sequence alignment (MSA) tool”, please delete “which is a progressive alignment technique”. Please delete the “multiple sequence alignment. the abbreviation “MSA” could be explain for the first time, and then author could use the “MSA” only.

Method :

It is too verbose, author do not need to explain advantage of every software or tools used. They have no relationship with your paper. Just give clear method.

Results:

Result was not consists of describe what is figure and table, rather give a results and explain using Figure and Table. Please re-write the results clearly.

Line 184-186: what sentence “only coding regions are considered for identification of hotspot mutations as the non-coding regions exhibit high entropy values and can be misleading while selecting such mutation points as hotspot mutations”?

Disccussion:

Line 21-26: The sentence “multiple sequence alignment of 5734 genomic sequences are carried out using MAFFT” . The author just needs to discuss the results rather than mention the method here.

Line 217-218: delete sentence “the details of which are already discussed in the Result Section”.

Conclusion:

The author does not need to describe the method and results again, just give the summary and discovery of this paper.

6. PLOS authors have the option to publish the peer review history of their article (what does this mean?). If published, this will include your full peer review and any attached files.

Reviewer #1: No

Reviewer #2: No

Reviewer #3: No

---

## [Author Response · Author response to Decision Letter 0]

8 Nov 2021

Reviewer #1: SARS-CoV-2 is still spreading around the world very rapidly with a high infection rate that could become a global pandemic. In order for researchers, including us, to design a more effective vaccine, we are analyzing the evolving virus strains.

You have performed multiple sequence alignments of 5734 sequences of SARS-CoV-2 using MAFFT and phylogenetic analysis using Nextstrain. Then, we identified several SNPs and point mutations. You identified the top 10 hotspot mutation points in the coding region. As a result, 130 hotspot mutations were identified in the temporal analysis and 250 in the spatial analysis. Subsequently, 32 temporally unique and 63 spatially unique hotspot mutations were identified, respectively. In addition, you have identified 21 point mutations in the time series analysis. For example, A97V in RdRp, L126F in NSP16, Q57H in ORF3a, and R203K, R203M, and G204R in Nucleocapsid. You also reaffirmed that the mutations of concern are E484Q and E484K in Spike.

minor issues

1. I could not read Tables 1 through 3 because the text was too small. Therefore, you should replace them with complete and readable tables in the text. Please move this table to the supplement.

Answer: We would like to apologise for the inconvenience caused. According to the suggestion, the tables have been modified in the revised manuscript. However, since these are very important tables, we have kept this in the main paper for the revised mauscript.

I believe that your paper needs to be released to the world as soon as possible.

The authors would like to thank the reviewer for the very kind comments.

Reviewer #2: This paper analyzes GISAID data to identify mutational hotspots within SARS-CoV-2 genomes sequenced in India. This analysis identifies several mutations that appear to change over time or vary between regions within India. The authors claim that these locations are mutational hotspots but do not provide compelling evidence to support this.

Answer: Mutations like L452R and P681R which are part of the Delta variant are also two of the hotspot mutations as identified by the analysis. It is to be noted that Delta variant was responsible for the catastrophic 2nd wave in India. Figures 10 (a) and (b) in the revised manuscript show the plot of confirmed and deceased cases in India till 31st October 2021. As can be seen from both the figures, western part of India has a very high number of confirmed and deceased cases which can be attributed to the Delta variant. As is shown in Table 2, Maharashtra which lies in the western part of India has both of the aforementioned mutations identified as hotspots. Also, some mutational hotspots are part of the Alpha, Beta and Delta variants as shown in Figure 9 in the revised manuscript, thereby confirming that they are indeed qualified to be hotspot mutations. These facts are elaborately discussed in the revised manuscript as well.

Specific comments:

1. Analysis of mutational hotspots is confounded by the competition between variants. Specifically, when one variant displaces another in a state or over time this will appear to show an enrichment for mutations associated with this variant even if the location of these mutations is not functionally important. At least for the positions identified as mutation “hotspots”, the authors should test whether they are changing in frequency within specific lineages. As an example, S:E484Q within the Delta lineage was presumed to be particularly problematic since it resembles the S:E484K mutation found in the Beta variant but it has since declined in frequency within the Delta variant.

Answer: According to the suggestion of the reviewer, the entropy change in hotspot mutations in variants like Alpha, Beta and Delta are reported in Figure 9 in the revised manuscript in order to illustrate the point as mentioned in the comment.

2. Please avoid the use of the term “double mutant” as it is scientifically misleading. Please refer to the strains by either their PANGO lineage (e.g. B.1.617.2) or by their WHO designation (e.g. Delta variant).

Answer: According to the suggestion of the reviewer, the changes have been made in the revised manuscript.

3. Figure captions need more description.

Answer: According to the suggestion of the reviewer, more descriptions have been added to the figures in the revised manuscript.

4. Variants refers to the combination of many mutations rather than any specific mutation. E.g. Page 2, line 93 is incorrect since E484Q is not a variant. Please fix terminology throughout.

Answer: According to the suggestion of the reviewer, the terminology has been fixed in the revised manuscript.

5. The protein structure model shown for nucleocapsid indicates a mostly unstructured architecture, but this is misleading. The N-terminal and C-terminal domain structures have been solved by multiple research groups and are known to be well ordered. Please update with a revised structure.

Answer: According to the suggestion of the reviewer, the structure of Nucleocapsid has been updated in the revised manuscript and the N-terminal and C-terminal have been confirmed using the PDBs 6M3M (range:50-174) and 6YUN (range:249-364) respectively. The following is the structure that has been used in the revised manuscript.

 6. The use of PROVEAN and similar tools to detect “damaging” mutations is not explained well and is potentially misleading. These tools were designed to detect the impact of mutations of human proteins rather than viral proteins with the assumption that major changes to human proteins are likely to be deleterious. This cannot be assumed for emerging viruses because they are under a rapidly changing selection conditions and mutations identified by PROVEAN might be beneficial for the virus to avoid immunity or even to enhance function. This needs to be discussed more clearly. It is also unclear how this helps to identify a location as a potential mutation hotspot.

Answer: It is important to note that in case of protein, damaging mostly defines instability. Generally, this is used for human proteins. As a consequence, if the human protein is damaging in nature because of mutations, then the human protein-protein interactions may occur with high or low binding affinity. Now in case of virus, similar consequences may happen which means if the virus protein is damaged because of mutations, it may interact with human proteins with similar binding affinity. As a result, the virus may acquire characteristics like transmissibility, escaping antibodies, etc. This is now clearly mentioned in the revised manuscript as well in order to avoid the confusion pertaining to the meaning of ‘damaging’ as concluded by PROVEAN and Polyphen-2. Moreover, we agree that the effects of these characteristics on human hosts are a matter of further investigations. Therefore, to draw a clear biological conclusion from the point of view of host, help of virologists is needed and as a future scope we are working in that direction. This is mentioned in the revised manuscript in the conclusion section.

Please note that hotspot mutations are characterized by PROVEAN, Polyphen-2 and I-mutant 2.0 after their locations have been identified by Nextstrain. Therefore, there is no relation between locations and the aforementioned tools. It is also to be noted that PROVEAN and Polyphen-2 are developed on more or less same background. Thus, their results are analogous to each other. Therefore, to avoid redundancy only the results from well-known tool Polyphen-2 are kept in the revised manuscript.

Reviewer #3: Comments to the Author: The manuscript give a meaningful view point to the analysis of evolving virus strains of SARS-CoV-2, but the paper is not clearly written.

Major points:

The paper from introduction to discussion should be simplified; it’s too long and too verbose. The manuscript did not give a meaningful view about what authors do this work and what they get conclusion. I recommend author to revise the manuscript as clear as possible.

1. Abstract: I strongly recommend author to re-write the abstract and give a clear abstract.

Answer: According to the suggestion of the reviewer, the abstract has been rewritten in the revised manuscript.

2. Introduction: It’s too verbose. It present too much description to others research. The author may likely just put others researches together rather than summary and conclude their studies.

Answer: According to the suggestion of the reviewer, the Introduction has been modified in the revised manuscript.

Line。。。： 300K should be replaced using a formal description, e.g 300,000. The problem is through the entire manuscript.

Answer: According to the suggestion of the reviewer, the changes have been made in the revised manuscript.

Line 10: The prevalent variant in South African is B.1.351, so “501Y. V2” should change to B.1.351.

Answer: According to the suggestion of the reviewer, the change has been made in the revised manuscript.

Line 11: Japanese should be Japan. Brazilian should be Brazil.

Answer: It is to be noted that in the revised manuscript, the variants of concern with the corresponding W.H.O declared naming conventions have been provided.

Line 11: E484K is not a linage, please clarify update the mainly variants name.

Answer: According to the suggestion of the reviewer, the change has been made in the revised manuscript.

Line 24: In [8] and In [9] should be replace as more reasonable description, It can be change by author name or change to another description. This problem is present through the whole paper.

Answer: According to the suggestion of the reviewer, the changes have been made in the revised manuscript.

Line 23-25: what this sentence relationship with previous viewpoint?

Answer: This is to be noted that this sentence was written inadvertently. We deeply apologise for this. The required change has been done in the revised manuscript.

Line 26-31: The author would like to

Answer: It is not very clear which sentence the reviewer is mentioning.

Line 31-32: what this sentence mean “thereby indicating potential impacts on the ongoing development of various COVID-19 diagnosis and cure”? and what this sentence relationship with these variants?

Answer: According to the suggestion of the reviewer, this sentence has been modified in the revised manuscript. The sentence indicates that Nucleocapsid cannot be a possible diagnostic target as it exhibits quite high number of mutations. Thus, this may undermine the ongoing researches targeting Nucleocapsid for COVID-19 diagnosis, vaccines, antibody and small-molecular drugs.

Line 36-37: previous have mentioned that “they have found Nucleocapsid to have the highest mutational changes in frequency”, the author can summary them together rather than describe it again and again.

Answer: According to the suggestion of the reviewer, the changes have been made in the revised manuscript.

Line 15-72: All citation view were list, please summary and clarify the paragraph.

Answer: According to the suggestion of the reviewer, the change has been made in the revised manuscript.

Line 75: the method “multiple alignment using fast fourier transform (MAFFT)” could move to method rather introduction.

Answer: It is to be noted that the only the method name has been mentioned in the Introduction to give the readers an overview. 

Line 77-94: The sentence about method should be move to “Material and Methods” part, the sentence about the result details should be move to Results part. The introduction just retain the summary of results and meaning of this paper.

Answer: According to the suggestion of the reviewer, the changes have been made in the revised manuscript.

Line 105: The citation is a lab? And the reference list 4 was not contain an author named Zhang.

Answer: It is to be noted that Zhang Lab is not a citation but a footnote to highlight the website from which the SARS-CoV-2 protein PDBs are collected. That is why the reference [4] (which is cited in the last line of the 1st paragraph of the Introduction) does not contain an author named Zhang.

Line 115: “MAFFT which is a progressive alignment technique is used as the multiple sequence alignment (MSA) tool”, please delete “which is a progressive alignment technique”. Please delete the “multiple sequence alignment. the abbreviation “MSA” could be explain for the first time, and then author could use the “MSA” only.

Answer: According to the suggestion of the reviewer, the changes have been made in the revised manuscript.

Method

 :

It is too verbose, author do not need to explain advantage of every software or tools used. They have no relationship with your paper. Just give clear method.

Answer: According to the suggestion of the reviewer, the changes have been made in the revised manuscript.

Results:

Result was not consists of describe what is figure and table, rather give a results and explain using Figure and Table. Please re-write the results clearly.

Answer: According to the suggestion of the reviewer, the changes have been made to the best of abilities and readers point of view in the revised manuscript.

Line 184-186: what sentence “only coding regions are considered for identification of hotspot mutations as the non-coding regions exhibit high entropy values and can be misleading while selecting such mutation points as hotspot mutations”?

Answer: Non-coding regions do not produce any protein to bind with human proteins. Thus, they are not considered for hotpot mutations as we have confined our research to only the coding regions. Moreover, in non-coding regions, the entropy value is high for almost all mutation points even if they may not be very important mutation points for SARS-CoV-2. Thus, instead of considering the mutation points in both coding and non-coding regions, we have only considered the mutation points in coding regions so that they exhibit the true characteristics of hotspot mutations.

Disccussion:

Line 21-26: The sentence “multiple sequence alignment of 5734 genomic sequences are carried out using MAFFT” . The author just needs to discuss the results rather than mention the method here.

Answer: According to the suggestion of the reviewer, the changes have been made in the revised manuscript. It is to be noted that instead of 5734 Indian SARS-CoV-2 genomes, the revised manuscript contains the analysis on 17271 such genomes as the number of genomes have updated over time. We have conducted all the experiments all over again to provide the updated results in the revised manuscript.

Line 217-218: delete sentence “the details of which are already discussed in the Result Section”.

Answer: According to the suggestion of the reviewer, the changes have been made in the revised manuscript.

Conclusion:

The author does not need to describe the method and results again, just give the summary and discovery of this paper.

Answer: According to the suggestion of the reviewer, the changes have been made in the revised manuscript.

---

## [Decision Letter · Decision Letter 1]

7 Mar 2022

Phylogenetic Analysis of 17271 Indian SARS-CoV-2 Genomes to Identify Temporal and Spatial Hotspot Mutations

PONE-D-21-17506R1

Dear Dr. Ghosh,

We’re pleased to inform you that your manuscript has been judged scientifically suitable for publication and will be formally accepted for publication once it meets all outstanding technical requirements.

Kind regards,

Chandrabose Selvaraj, Ph.D.

Academic Editor

PLOS ONE

Additional Editor Comments (optional):

Reviewers' comments:

Reviewer's Responses to Questions

**Comments to the Author**

1. If the authors have adequately addressed your comments raised in a previous round of review and you feel that this manuscript is now acceptable for publication, you may indicate that here to bypass the “Comments to the Author” section, enter your conflict of interest statement in the “Confidential to Editor” section, and submit your "Accept" recommendation.

Reviewer #3: All comments have been addressed

Reviewer #4: All comments have been addressed

2. Is the manuscript technically sound, and do the data support the conclusions?

Reviewer #3: Yes

Reviewer #4: Yes

3. Has the statistical analysis been performed appropriately and rigorously? 

Reviewer #3: Yes

Reviewer #4: N/A

4. Have the authors made all data underlying the findings in their manuscript fully available?

Reviewer #3: Yes

Reviewer #4: Yes

5. Is the manuscript presented in an intelligible fashion and written in standard English?

Reviewer #3: Yes

Reviewer #4: Yes

6. Review Comments to the Author

Reviewer #3: Author has addressed the issues that I mentioned.I believe that this paper needs to be released to the world as soon as possible.

Reviewer #4: The authors answered all questions from reviewers and made all changes to the manuscript, which can be accepted in this format.

7. PLOS authors have the option to publish the peer review history of their article (what does this mean?). If published, this will include your full peer review and any attached files.

Reviewer #3: No

Reviewer #4: **Yes: **Fabrício Souza Campos

---

## [Editor Report · Acceptance letter]

18 Mar 2022

PONE-D-21-17506R1 

Phylogenetic Analysis of 17271 Indian SARS-CoV-2 Genomes to Identify Temporal and Spatial Hotspot Mutations 

Dear Dr. Saha:

I'm pleased to inform you that your manuscript has been deemed suitable for publication in PLOS ONE. Congratulations! Your manuscript is now with our production department. 

Kind regards, 

on behalf of

Dr. Chandrabose Selvaraj 

Academic Editor

PLOS ONE